# HyperDynamics: Meta-Learning Object and Agent Dynamics with Hypernetworks

**Zhou Xian, Shamit Lal, Hsiao-Yu Fish Tung, Emmanouil Antonios Platanios & Katerina Fragkiadaki**
School of Computer Science, Carnegie Mellon University, Pittsburgh, PA 15213, USA
{xianz1,shamitl,htung,e.a.platanios,katef}@cs.cmu.edu

## Abstract

We propose HyperDynamics, a dynamics meta-learning framework that conditions on an agent's interactions with the environment and optionally its visual observations, and generates the parameters of neural dynamics models based on inferred properties of the dynamical system. Physical and visual properties of the environment that are not part of the low-dimensional state yet affect its temporal dynamics are inferred from the interaction history and visual observations, and are implicitly captured in the generated parameters. We test HyperDynamics on a set of object pushing and locomotion tasks. It outperforms existing dynamics models in the literature that adapt to environment variations by learning dynamics over high dimensional visual observations, capturing the interactions of the agent in recurrent state representations, or using gradient-based meta-optimization. We also show our method matches the performance of an ensemble of separately trained experts, while also being able to generalize well to unseen environment variations at test time. We attribute its good performance to the multiplicative interactions between the inferred system properties—captured in the generated parameters—and the low-dimensional state representation of the dynamical system.

## 1 Introduction

Humans learn dynamics models that predict results of their interactions with the environment, and use such predictions for selecting actions to achieve intended goals (Miall & Wolpert, 1996; Haruno et al., 1999). These models capture intuitive physics and mechanics of the world and are remarkably versatile: they are expressive and can be applied to all kinds of environments that we encounter in our daily lives, with varying dynamics and diverse visual and physical properties. In addition, humans do not consider these models fixed over the course of interaction; we observe how the environment behaves in response to our actions and quickly adapt our model for the situation at hand based on new observations. Let us consider the scenario of moving an object on the ground. We can infer how heavy the object is by simply looking at it, and we can then decide how hard to push. If it does not move as much as expected, we might realize it is heavier than we thought and increase the force we apply (Hamrick et al., 2011).

Motivated by this, we propose *HyperDynamics*, a dynamics meta-learning framework for that generates parameters for dynamics models (*experts*) dedicated to the situation at hand, based on observations of how the environment behaves. HyperDynamics has three main modules: i) an encoding module that encodes a few agent-environment interactions and the agent's visual observations into a latent feature code, which captures the properties of the dynamical system, ii) a hypernetwork (Ha et al., 2016) that conditions on the latent feature code and generates parameters of a dynamics model dedicated to the observed system, and iii) a target dynamics model constructed using the generated parameters that takes as input the current low-dimensional system state and the agent action, and predicts the next system state, as shown in Figure 1. We will be referring to this target dynamics model as an *expert*, as it specializes on encoding the dynamics of a particular scene at a certain point in time. HyperDynamics conditions on real-time observations and generates dedicated expert models on the fly. It can be trained in an end-to-end differentiable manner to minimize state prediction error of the generated experts in each task.

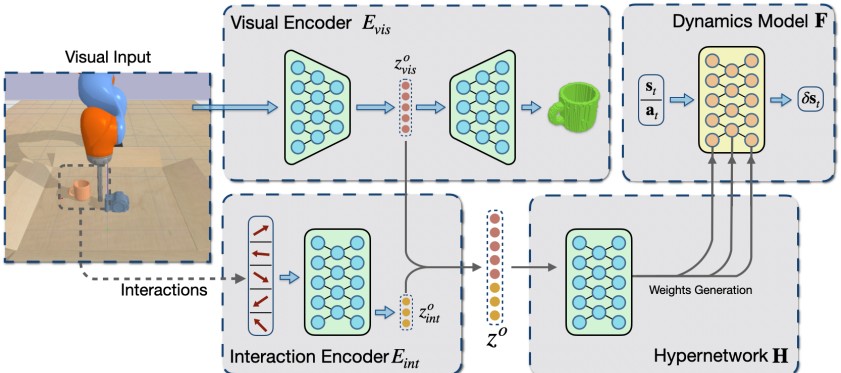

**Figure 1:** HyperDynamics encodes the visual observations and a set of agent-environment interactions and generates the parameters of a dynamics model dedicated to the current environment and timestep using a hypernetwork. HyperDynamics for pushing follows the general formulation, with a visual encoder for detecting the object in 3D and encoding its shape, and an interaction encoder for encoding a small history of interactions of the agent with the environment.

Many contemporary dynamics learning approaches assume a fixed system without considering potential change in the underlying dynamics, and train a fixed dynamics model (Watter et al., 2015; Banijamali et al., 2018; Fragkiadaki et al., 2015; Zhang et al., 2019). Such expert models tend to fail when the system behavior changes. A natural solution to address this is to train a separate expert for each dynamical system. However, this can hardly scale and doesn't transfer to systems with novel properties. Inspired by this need, HyperDynamics aims to infer a system's properties by simply observing how it behaves, and automatically generate an expert model that is dedicated to the observed system. In order to address system variations and obtain generalizable dynamics models, many other prior works propose to condition on visual inputs and encode history of interactions to capture the physical properties of the system (Finn & Levine, 2016; Ebert et al., 2018; Xu et al., 2019b; Pathak et al., 2017; Xu et al., 2019a; Li et al., 2018; Sanchez-Gonzalez et al., 2018b; Hafner et al., 2019). These methods attempt to infer system properties; yet, they optimize a single and fixed global dynamics model, which takes as input both static system properties and fast-changing states, with the hope that such a model can handle variations of systems and generalize. We argue that a representation which encodes system information varies across different system variations, and each instance of this presentation ideally should correspond to a different dynamics model that needs to be used in the corresponding setting. These models are different, but also share a lot of information as similar systems will have similar dynamics functions. HyperDynamics makes explicit assumptions about the relationships between these systems, and attempts to exploit this regularity and learn such commonalities across different system variations. There also exsists a family of approaches that attempt online model adaptation via meta-learning (Finn et al., 2017; Nagabandi et al., 2019; Clavera et al., 2018; Nagabandi et al., 2018a). These methods perform online adaptation on the parameters of the dynamics models through gradient descent. In this work, we empirically show that our approach adapts better than such methods to unseen system variations.

We evaluate HyperDynamics on single- and multi-step state predictions, as well as downstream model-based control tasks. Specifically, we apply it in a series of object pushing and locomotion tasks. Our experiments show that HyperDynamics is able to generate performant dynamics models that match the performance of separately and directly trained experts, while also enabling effective generalization to systems with novel properties in a few-shot manner. We attribute its good performance to the multiplicative way of combining explicit factorization of encoded system features with the low-dimensional state representation of the system. In summary, our contributions are as follows:

- We propose HyperDynamics, a general dynamics learning framework that is able to generate expert dynamics models on the fly, conditioned on system properties.

- We apply our method to the contexts of object pushing and locomotion, and demonstrate that it matches performance of separately trained system-specific experts.

- We show that our method generalizes well to systems with novel properties, outperforming contemporary methods that either optimize a single global model, or attempt online model adaptation via meta-learning.

## 2 RELATED WORK

### 2.1 MODEL LEARNING AND MODEL ADAPTATION

To place HyperDynamics in the context of existing literature on model learning and adaptation, we distinguish four groups of methods, depending on whether they explicitly distinguish between dynamic—such as joint configurations of an agent or poses of objects being manipulated—and static —such as mass, inertia, 3D object shape—properties of the system, and whether they update the dynamics model parameters or the static property values of the system.

**(i)** No adaptation. These methods concern dynamics model over either low-dimensional states or high-dimensional visual inputs of a specific system, without considering potential changes in the underlying dynamics. These models tend to fail when the system behavior changes. Such *expert* dynamics models are popular formulations in the literature (Watter et al., 2015; Banijamali et al., 2018; Fragkiadaki et al., 2015; Zhang et al., 2019). They can be adapted through gradient descent at any test system, yet that would require a large number of interactions.

**(ii)** Visual dynamics and recurrent state representations. These methods operate over high-dimensional visual observations of systems with a variety of properties (Finn & Levine, 2016; Ebert et al., 2018; Li et al., 2018; Xu et al., 2019b; Mathieu et al., 2015; Oh et al., 2015; Pathak et al., 2017), hoping the visual inputs could capture both the state and properties of the system. Some also attempt to encode a history of interactions with a recurrent hidden state (Xu et al., 2019a; Li et al., 2018; Ebert et al., 2018; Sanchez-Gonzalez et al., 2018b; Hafner et al., 2019), in order to implicitly capture information regarding the physical properties of the system. These methods use a single and fixed global dynamics model that takes system properties as input directly, together with its state and action.

**(iii)** System identification (Bhat et al., 2002). These methods model explicitly the properties of the dynamical system using a small set of semantically meaningful physics parameters, such as mass, inertia, damping, etc. (Ajay et al., 2019; Mrowca et al., 2018), and adapt by learning their values from data on-the-fly (Chen et al., 2018; Li et al., 2019; Sanchez-Gonzalez et al., 2018b; Battaglia et al., 2016; Sanchez-Gonzalez et al., 2018a).

**(iv)** Online model adaptation via meta-learning (Finn et al., 2017; Nagabandi et al., 2019; Clavera et al., 2018; Nagabandi et al., 2018a). These methods perform online adaptation on the parameters of the dynamics models through gradient descent. They are optimized to produce a good parameter initialization that can be adapted to the local context with only a few gradient steps at test time.

The expert models in (i) work well for their corresponding systems but fail to handle system variations. Methods in (ii) and (iii), though attempting to infer system properties, optimize a single and fixed global dynamics model, which takes as input both static system properties and fast-changing states. Methods in (iv) attempt online update for a learned dynamics model. HyperDynamics differs from these methods in that it generates system-conditioned dynamics models on the fly. In our experimental Section, we compare with all these methods and empirically show that our method adapts better to unseen systems.

### 2.2 BACKGROUND ON HYPERNETWORKS

Central to HyperDynamics is the notion of using one network (the hypernetwork) to generate the weights of another network (the target network), conditioned on some forms of embeddings. The weights of the target network are not model parameters; during training, their gradients are back-propagated to the weights of the hypernetwork (Chang et al., 2019). This idea is initially proposed in (Ha et al., 2016), where the authors demonstrate leveraging the structural features of the original network to generate network layers, and achieve model compression while maintaining competitive performance for both image recognition and language modelling tasks. Since then, it has been applied to various domains. In Ratzlaff & Fuxin (2019) the authors use hypernetworks to generate an ensemble of networks for the same task to improve robustness to adversarial data. Platanios et al. (2018) attempts to generate language encoders conditioned on the language context for better weight sharing across languages in a machine translation setting. Other applications include but are not limited to: weight pruning (Liu et al., 2019), multi-task learning (Klocek et al., 2019; Serrà et al., 2019; Meyerson & Miikkulainen, 2019), neural architecture search (Zhang et al., 2018; Brock et al.,

2017), and continual learning (von Oswald et al., 2019). To the best of our knowledge, this is the first work that applies the idea of generating model parameters to the domain of model-based RL, and proposes to generate expert models conditioned on system properties.

## 3 HYPERDYNAMICS

HyperDynamics generates expert dynamics models on the fly, conditioned on observed system features. We study applying HyperDynamics to a series of object pushing and locomotion tasks, but in principle it is agnostic to the specific context of a dynamics learning task. In this section, we first present the general form of HyperDynamics, and then describe the task-specific modules we used.

Given a dynamical system $o$ (which is typically composed of an agent and its external environment) with a certain set of properties, HyperDynamics observes both how it behaves under a few actions (interactions), as well as its visual appearance when needed. Then, a visual encoder $E_{vis}$ encodes the visual observation $I$ into a latent visual feature code $z_{vis}^o$, and an interaction encoder $E_{int}$ encodes a $k$-step interaction trajectory $\tau = (s_0, a_0, ...s_k, a_k)$ into a latent physics feature code $z_{int}^o$. We then combine these two features codes by concatenating them, $z^o = [z_{int}, z_{vis}]$, forming a single vector that captures the underlying dynamics properties of the system. A *hypernetwork* $\mathbf{H}$ maps $z^o$ to a set of neural network weights for a forward dynamics model $\mathbf{F}^o$, which is a mapping from the state of the system and the action of the agent, to the state change at the subsequent time step $t + 1$: $\delta s_t = \mathbf{F}^o(s_t, a_t)$. The hypernetwork is a fully-connected network and generates all parameters in a single-shot as the output of it's final layer. The detailed architecture of HyperDynamics for an object pushing example is shown in Figure 1.

We meta-train HyperDynamics by backpropagating error gradients of the generated dynamics model's prediction, all the way to our visual and interaction encoders and the hypernetwork. Let $\mathcal{O}$ denote the pool of all systems used for training. Let $T$ denote the length of the collected trajectories. Let $N$ denote the number of trajectories collected per system. The final prediction loss for our model is defined as:

$$\mathcal{L}_{pred} = \frac{1}{|\mathcal{O}|NT} \sum_{o=1}^{|\mathcal{O}|} \sum_{i=1}^{N} \sum_{t=1}^{T} \|\delta s_t - \mathbf{F}(s_t, a_t; \mathbf{H}([E_{vis}(I; \phi), E_{int}(\tau; \psi)]; \omega))\|_2, \quad (1)$$

where $\phi$, $\psi$ and $\omega$ denote the learned parameters of $E_{vis}$, $E_{int}$ and $\mathbf{H}$, respectively. We consider a set of locomotion tasks for which only the interaction history is used as input, in accordance with prior work (e.g., Finn et al., 2017; Nagabandi et al., 2019; Clavera et al., 2018; Nagabandi et al., 2018a), and a pushing task on a diverse set of objects, where the visual encoder encodes the 3D shape of the object.

### 3.1 HYPERDYNAMICS FOR OBJECT PUSHING

For our object pushing setting, we consider a robotic agent equipped with an RGB-D camera to observe the world and an end-effector to interact with the objects. The agent's goal is to push objects to desired locations on a table surface. At time $t$, the state of our pushing system is $s_t^o = < s_t^{obj}, s_t^{rob} >$, where the object state $s_t^{obj}$ is a 13-vector containing the object's position $\mathbf{p}_t^{obj}$, orientation $\mathbf{q}_t^{obj}$ (represented as a quaternion), as well as its translational and rotational velocities $\mathbf{v}_t^{obj}$, and the robot state $s_t^{rob}$ is a 3-vector representing the position of its end-effector. For controlling the robot, we use position-control in our experiments, so the action $a_t = (\delta x, \delta y, \delta z)$ is a 3-vector representing the movement of the robot's end-effector.

Objects behave differently under pushing due to varying properties such as shape, mass, and friction relative to the table surface. For this task, HyperDynamics tries to infer object properties from its observations and generate a dynamics model specific to the object being manipulated. While the 3D object shape can be inferred by visual observations, physics properties such as mass and friction are hard to discern visually. Instead, they can be inferred by observing the motion of an object as a result to a few interactions with the robot (Xu et al., 2019b). To produce the code $z^o$, we use both $E_{vis}$ for encoding shape information and $E_{int}$ for encoding physics properties.

Given an object to be pushed, we assume its properties stay unchanged during the course of manipulation. To produce the interaction trajectory $\tau$, the robot pushes the object by $k$ actions sampled from

random directions. $E_{int}$ then maps $\tau$ to a latent code $z_{int} \in \mathbb{R}^2$. In practice we found $k = 5$ sufficient to yield informative $z_{int}$ for pushing, and using a bigger value does not help further improve performance. We use a simple fully-connected network for $E_{int}$.

Given a single RGB-D image $I$ of the scene, $E_{vis}$ produces object-centric $z_{vis} \in \mathbb{R}^8$ to encode the shape information of the objects in the scene. We build our $E_{vis}$ using the recently proposed Geometry-Aware Recurrent Networks (GRNNs) (Tung et al., 2019), which learn to map single 2.5D images to complete 3D feature grids $\mathbf{M} \in \mathbb{R}^{w \times h \times d \times c}$ that represent the appearance and the shape of the scene and its objects. GRNNs learn such mappings using differentiable unprojection (2.5D to 3D) and projection (3D to 2.5D) neural modules and can be optimized end-to-end for both 3D object detection and view prediction, and complete missing or occluded shape information while optimized for such objectives. It helps us implicitly handle possible minor occlusions caused by the robot end-effector during data collection. GRNNs detect objects using a 3D object detector that operates over its latent 3D feature map $\mathbf{M}$ of the scene. The object detector maps the scene feature map $\mathbf{M}$ to a variable number of axis-aligned 3D bounding boxes of the objects (and their segmentation masks if needed). Its architecture is similar to Mask R-CNN (He et al., 2017) but it uses RGB-D inputs and produces 3D bounding boxes. We refer readers to Tung et al. (2019) for more details.

$E_{vis}$ first detects all the objects in the scene, and then crops the scene map around the object we want to push to produce an object-centric 3D feature map $\mathbf{M}^{obj} = \text{crop}(\mathbf{M})$, which is then used for shape encoding. We opt for such 3D representations since it allows a more realistic pushing setting: we do not require the camera viewpoint to be fixed throughout training and testing, and we can handle robot-object occlusions better since our encoder learns to complete missing (occluded) shape information in the map $\mathbf{M}^{obj}$. We train $E_{vis}$ for object detection jointly with the prediction loss in Equation 1. We also add additional regularization losses by reconstructing the object's shape, where the visual code $z_{vis}$ is passed into a decoder $D_{vis}$ to reconstruct the object shape $V^{obj}$. Note that the reconstruction loss can be applied from pure visual data, by observing objects from multiple viewpoints, without necessarily interacting with them. This is important because moving a camera around objects is cheaper than interacting with them through the robot's end-effectors. In addition, since our tasks concern planar pushing, we found reconstructing a top-down 2D shape instead of the the full 3D one suffices for regularization, essentially predicting a top-down view of the object from any perspective viewpoint.

**Object 3D orientation: context or state?** The shape of rigid objects does not change over time; rather, its orientation changes. The framework discussed so far intuitively divides the input information among different modules: the hypernetwork generates a dynamics model conditioned on object properties, and the dynamics model takes care of low-dimensional object states. Then, one natural choice is that the object state includes both its position and orientation information, and $z_{vis}$ encodes the object shape information oriented at its canonical orientation, independent of its actual orientation. However, when there's only a few objects used for training, the hypernetwork will only strive to optimize for a discrete set of dynamics models, as it only sees a discrete set of object-specific $z_{vis}$, thus potentially leading to severe overfitting on the training data. Instead of producing $z_{vis}$ to encode object shape information at its canonical pose, we propose to encode the object shape oriented at its current orientation at each time step, and discard the orientation information from the states fed into the generated dynamics model, as their generated parameters would already encode the orientation information. This way, we move the orientation information from the state input (fed into the generated dynamics model) to the shape input (encoded into $z_{vis}$ and fed into the hypernetwork $\mathbf{H}$). Using oriented as opposed to canonical shape as input permits $\mathbf{H}$ to have way more training examples, even from a small number of objects with distinct shapes. Intuitively, $\mathbf{H}$ now treats the same object with different orientations as distinct objects, and generates a different expert for each. This helps smoothen and widen the data distribution seen by $\mathbf{H}$.

## 3.2 HyperDynamics for Locomotion

For locomotion tasks, we consider a legged robot walking in a dynamic environment with changing terrains, where their properties are only locally consistent. At timestep $t$, the state contains the joint velocities and joint positions of the robot, as well as the position of its center of mass. The dimension of the state depends on the actual morphology of the robot. Torque-control is used to send low-level torque commands to the robot.

Unlike the object pushing setting where the properties of the objects can be safely assumed to remain constant, here the system dynamics changes rapidly due to varying terrain properties. Hence, we adopt an online adaptation setting where at timestep $t$, the interaction data is taken to be the most recent trajectory $\tau = (s_{t-k}, a_{t-k}, ...s_{t-1}, a_{t-1})$. Similarly, we use a simple fully-connected network to encode the interaction trajectory. In practice, we found $k = 16$ (equal to a trajectory segment of around 0.3 seconds) to be informative enough for $E_{int}$ to infer accurate terrain properties, and using a bigger value does not provide further performance gain. The encoded $z_{int} \in \mathbb{R}^2$ is then used by $\mathbf{H}$ to generate a dynamics model for the local segment of the terrain in real-time.

### 3.3 Model Unrolling and Action Planning

Action-conditioned dynamics models can be unrolled forward in time for long-term planning and control tasks. We apply our approach with model predictive control (MPC) for action planning and control. Specifically, we use random shooting (RS) (Nagabandi et al., 2018b) for action planning, unroll our model forward with randomly sampled action sequences, pick the best performing one, and replan at every timestep using updated state information to avoid compounding errors.

The unrolling mechanism of HyperDynamics for pushing is straightforward: at each timestep $t$, a generated dynamics model $\mathbf{F}_t^o$ predicts $\delta s_t^o$ for the system. Afterwards, we compute $s_{t+1}^o = s_t^o \oplus \delta s_t^o$, where $\oplus$ represents simple summation for positions and velocities, and quaternion composition for orientations. We then re-orient the object-centric feature map $\mathbf{M}^{obj}$ using the updated object orientation $\mathbf{q}_{t+1}^{obj}$ and generate a new dynamics model $\mathbf{F}_{t+1}^o$. Finally, the updated full state $s_{t+1}^o$, excluding $\mathbf{q}_{t+1}^{obj}$, is used by $\mathbf{F}_{t+1}^o$ for motion prediction. For locomotion, at each timestep we obtain one $z_{int}$, generate a dynamics model $\mathbf{F}_t^o$ for the local context, and use it for all unrolling steps into the future, since future terrain properties are not predictable.

## 4 Experiments

Our experiments aim to answer these questions: (1) Is HyperDynamics able to generate dynamics models across environment variations that perform as well as expert dynamics models, which are trained specifically on each environment variation? (2) Does HyperDynamics generalize to systems with novel properties? (3) How does HyperDynamics compare with methods that either use fixed global dynamics models or adapt their parameters during the course of interaction through meta-optimization? We test our proposed framework in the tasks of both object pushing and locomotion, and describe each of them in details below.

### 4.1 Object Pushing

Many prior works that learn object dynamics consider only quasi-static pushing or poking, where an object always starts to move or stops together with the robot's end-effector (Finn & Levine, 2016; Agrawal et al., 2016; Li et al., 2018; Pinto & Gupta, 2017). We go beyond simple quasi-static pushing by varying the physics parameters of the object and the scene, and allow an object to slide by itself if pushed with a high speed. A PID controller controls the robot's torque in its joint space under the hood, based on position-control commands, so varying the magnitude of the action could lead to different pushing speed of the end-effector. We test HyperDynamics on its motion prediction accuracy for single- and multi-step object pushing, as well as its performance when used for pushing objects to desired locations with MPC.

**Experimental Setup.** We train our model with data generated using the PyBullet simulator (Coumans & Bai, 2016–2019). Our setup uses a Kuka robotic arm equipped with a single rod-shaped end-effector, as shown in Figure 1. Our dataset consists of only 31 different object meshes with distinct shapes, including 11 objects from the MIT Push dataset (Yu et al., 2016) and 20 objects selected from four categories (*camera*, *mug*, *bowl* and *bed*) in the ShapeNet Dataset (Chang et al., 2015). We split our dataset so that 24 objects are used for training and 7 are used for testing. The objects can move freely on a planar table surface of size $0.6m \times 0.6m$. For data collection, we randomly sample one object with randomized mass and friction coefficient, and place it on the table with a random starting position. The mass is uniformly sampled from $[300, 1000]$, the friction coefficient of the object material is uniformly sampled from $[8e^{-4}, 12e^{-4}]$, and the friction coefficient of the table surface is set to be 10, all using the default units in PyBullet. Then, we instantiate the

**Table 1:** Motion prediction error (in centimeters).

| Model | Seen Objects | | Novel Objects | |
|---|---|---|---|---|
| | $t = 1$ | $t = 5$ | $t = 1$ | $t = 5$ |
| Expert-Ens | **0.82±0.37** | **4.22±2.21** | 1.68±0.79 | 5.83±2.49 |
| XYZ | 1.45±0.62 | 5.89±2.46 | 1.72±0.66 | 6.46±2.92 |
| Direct | 1.01±0.41 | 5.14±2.30 | 1.24±0.54 | 5.60±2.59 |
| MB-MAML | 1.68±0.61 | 8.91±3.79 | 1.76±0.73 | 9.05±5.98 |
| HyperDynamics | **0.83±0.35** | **4.27±2.24** | **0.93±0.53** | **4.77±2.57** |

**Table 2:** Pushing success rate.

| Model | Seen Objects | | Novel Objects | |
|---|---|---|---|---|
| | w/o obs | w/ obs | w/o obs | w/ obs |
| Expert-Ens | **0.92** | **0.72** | 0.80 | 0.44 |
| XYZ | 0.80 | 0.42 | 0.74 | 0.44 |
| Direct | 0.88 | 0.62 | 0.84 | 0.58 |
| MB-MAML | 0.70 | 0.42 | 0.68 | 0.38 |
| VF | 0.62 | — | 0.52 | — |
| DensePhysNet | 0.70 | — | 0.58 | — |
| HyperDynamics | **0.92** | **0.70** | **0.92** | **0.68** |

end-effector nearby the object and collect pushing sequences with length of 5 timesteps, where each action is a random horizontal displacement of the robot's end-effector ranging from $3cm$ to $6cm$, and each timestep is defined to be $800ms$. Our method relies on a single 2.5D image as input for object detection and motion prediction. Note that although collecting actual interactions with objects is expensive and time-consuming, data consisting of only varying shapes are cheap and easily accessible. To ensure $z_{vis}$ produced by the visual encoder $E_{vis}$ captures sufficient shape information for pushing, we train it on the whole ShapeNet (Chang et al., 2015) for shape autoencoding as an auxiliary task, jointly with the dynamics prediction task. See Appendix A.1 for more details.

**Baselines.** We compare our model against these baselines: **(1)** *XYZ*: a model that only take as input the action and states of the object and the agent, without having access to its visual feature or physics properties, similar to Andrychowicz et al. (2020) and Wu et al. (2017). **(2)** *Direct*: this baseline uses the same visual encoder and interaction encoder as our model does, and also uses the same input, while it passes the encoded latent code $z$ directly into a dynamics model in addition to the system state. Compared to ours, the only difference is that it directly optimizes a single global dynamics model instead of generating experts. This approach resembles the most common way of conditioning on object features used in the current literature (Finn & Levine, 2016; Agrawal et al., 2016; Li et al., 2018; Xu et al., 2019a; Fragkiadaki et al., 2015; Zhang et al., 2019; Hafner et al., 2019), where a global dynamics model takes as input both the state and properties of the dynamical system. **(3)** Visual Foresight (*VF*) (Ebert et al., 2018) and *DensePhysNet* (Xu et al., 2019a): these two models rely on 2D visual inputs of the current frame, and predict 2D optical flows that transform the current frame to the future frame. These models also use a global dynamics model for varying system properties, and feed system features directly into the model, following the same philosophy of *Direct*. These two baselines are implemented using the architectures described in their corresponding papers. **(4)** *MB-MAML* (Nagabandi et al., 2019): a meta-trained model which applies model-agnostic meta-learning to model-based RL, and trains a dynamics model that can be rapidly adapted to the local context when given newly observed data. This baseline also uses the same input as our model does; the only difference is that it uses the interaction data for online model update. **(5)** *Expert-Ens*: we train a separate expert model for each distinct object, forming an ensemble of experts. This baseline assumes access to the *ground-truth* orientation, mass, and friction, and serves as an oracle when tested on seen objects. For unseen objects, we select a corresponding expert from the ensemble using shape-based nearest-neighbour retrieval, essentially performing system identification.

**Implementation Details.** In order to ensure a fair comparison, all the dynamics models used in *Direct*, *MB-MAML* and *Expert-Ens* use the exactly same architecture as the one generated by HyperDynamics. The interaction encoder is a fully-connected network with one hidden layer of size 8. The hypernetwork is a fully-connected network with one hidden layer of size 16. The visual encoder follows the architecture of (Tung et al., 2019) and its encoder uses 3 convolutional layers, each followed by a max pooling layer, with a fully-connected layer at the end. It uses kernels of size 5, 3 and 3, with 2, 4, and 8 channels respectively. Its decoder uses 3 transposed convolutional layers, with 16, 8, and 1 channels and the same kernel size of 4. The generated dynamics model is a fully-connected network with 3 hidden layers, each of size 32; the same applies to the dynamics models used in *XYZ*, *Direct* and *Expert-Ens*. We use leaky ReLU as activations for all the modules. We use batch size of 8 and a learning rate of $1e - 3$. We collected $50,000$ pushing trajectories for training, and $1,000$ trajectories for testing. All models are trained till convergence for 500K steps.

We evaluate all the methods on prediction error for both single-step ($t = 1$) and multi-step unrolling ($t = 5$) (Table 1). We also test their performance for downstream manipulation tasks, where the robot needs to push the objects to desired target locations with MPC, in scenes both with and without obstacles. When obstacles are present, the agent needs to plan a collision-free path. We report their success rates in 50 trials in Table 2. Red numbers in bold denote the best performance across all

models, while black numbers in bold represent the oracle performance (provided by *Expert-Ens*). *VF* and *DensePhysNet* are not tested for motion prediction since they work in 2D image space and do not predict explicit object motions, and are not applicable for collision checking and collision-free pushing for the same reason.

Our model outperforms all baselines by a margin on both prediction and control tasks. When tested on objects seen during training, while our model needs to infer orientation information and physics properties via encoding visual input and interactions, it performs on par with the *Expert-Ens* oracle, which assumes access to ground-truth orientation, mass and friction coefficient. This shows that the dynamics model generated by HyperDynamics on the fly is as performant as the separately and directly trained dynamics experts. The *XYZ* baseline depicts the best performance possible when shape and physics properties of the objects are unknown. When tested with novel objects unseen during training, our model shows a performance similar to seen objects, outperforming *XYZ* by a great margin, demonstrating that our model transfers acquired knowledge to novel objects well. *Expert-Ens* on novel objects performs similarly to *XYZ*, suggesting that with only a handful of distinct objects in the training set, applying these dedicated models to novel objects, though after nearest-neighbour searches, yields poor generalization. Performance gain of our model over *Direct* suggests our hierarchical way of conditioning on system features and generating experts outperforms the common framework in the literature, which uses a global model with fixed parameters for varying system dynamics. Our model also shows a clear performance gain over *MB-MAML*, when only 5 data samples of interaction are available for online adaptation. This indicates that our framework, which directly generates dynamics model given the system properties, is more sample-efficient than the meta-trained dynamics model prior that needs to adapt to the context via extra tuning.

## 4.2 LOCOMOTION

**Experimental Setup.** We set up our environment using the Mu-JoCo simulator (Todorov et al., 2012). In particular, we consider two types of robots: a planar half-cheetah and a quadrupedal "ant" robot. For each robot, we consider two environments where the terrains are changing: (1) *Slope*: the robot needs to walk over a terrain consisting of multiple slopes with changing steepness. (2) *Pier*: the robot needs to walk over a series of blocks floating on the water surface, where damping and friction properties vary between the blocks. As a result, we consider these 4 tasks in total: *Cheetah-Slope*, *Ant-Slope*, *Cheetah-Pier* and *Ant-Pier*. Figure 2 shows two of them. Unlike the setting of object pushing, randomly and uniformly sampled states in the robot's state space do not match the actual state distribution encountered when the robot moves on the terrain. As a result, on-policy data collection is performed: the lat-

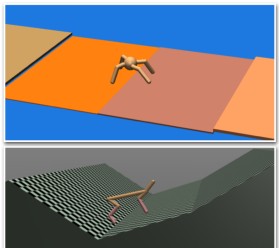

**Figure 2:** We consider two types of robots and two environments for locomotion. Top: Ant-Pier. Down: Cheetah-Slope.

est model is used for action selection to collect best-performing trajectories, which are then used to update the dataset for model training. For *Slope*, the environment is comprised of a series of consecutive terrain segments, each of which has a randomly sampled steepness. During test time, we also evaluate on unseen terrains with either higher or lower steepness. For *Pier*, each of the blocks floating on the water has randomly sampled friction and damping properties, and moves vertically when the robot runs over it, resulting in a rapidly changing terrain. Similarly, during test time we also evaluate using blocks with properties outside of the training distribution. See Appendix A.2 for more details.

**Baselines.** We compare our model against these baselines: **(1)** *Recurrent* (Sanchez-Gonzalez et al., 2018b): A recurrent model that also uses GRUs (Cho et al., 2014) to encode a history of interactions to infer system dynamics and perform system identification implicitly. **(2)** *MB-MAML* (Nagabandi et al., 2019): a model meta-trained using MAML, same as the one used for object pushing. It uses the same 16-step interaction trajectory for online adaptation. **(3)** *Expert-Ens*: An ensemble of separately trained experts, similar to the one used for object pushing. The only difference is that in pushing, we train one expert for each distinct object, while now we have a continuous distribution of terrain properties used for training. Therefore, for each task we pick several environments with properties evenly sampled from the continuous range, and train an expert for each of them.

**Implementation Details.** Again, in order to have a fair comparison, all the forward dynamics models used in *Recurrent* and *MB-MAML* use the same architecture as the one generated by Hy-

**Table 3:** Comparison of average total return for locomotion tasks.

| Model | Cheetah-Slope | | Cheetah-Pier | | Ant-Slope | | Ant-Pier | |
|---|---|---|---|---|---|---|---|---|
| | Seen | Novel | Seen | Novel | Seen | Novel | Seen | Novel |
| Expert-Ens | **104.1±59.2** | 76.9±42.4 | **354.6±182.9** | 321.4±171.3 | **142.1±77.9** | 124.2±71.9 | **219.0±113.2** | 179.3±87.2 |
| Recurrent | 64.4±21.0 | 76.6±38.1 | 292.5±132.1 | 279.7±132.4 | **133.9±68.4** | **142.1±66.1** | 176.4±87.7 | 170.5±89.1 |
| MB-MAML | 55.3±24.5 | 62.7±37.7 | 291.3±143.0 | 297.1±151.9 | 104.5±45.2 | 110.4 ±53.2 | 180.2±91.2 | 171.0±87.1 |
| HyperDynamics | **107.9±63.1** | **124.5±64.7** | **349.4±189.0** | **337.2±179.4** | **138.8±65.3** | **140.1±72.0** | **216.6±121.2** | **208.4±103.8** |

perDynamics. Both *Recurrent* and *MB-MAML* use the same input as our model does. Our encoder is a fully-connected network with 2 hidden layers, both of size 128, followed by ReLU activations and a final layer which outputs the encoded $z_{int}$. The hypernetwork is a fully-connected network with one hidden layer of size 16. The generated dynamics model is a fully-connected network with two hidden layers, each of size 128; the same applies to the dynamics models used in *Recurrent* and *MB-MAML*. During planning and control, 500 random action sequences are sampled with a planning horizon of 20 steps. Data collection is performed in an on-policy manner. We start the first iteration of data collection with randomly sampled actions and collect 10 rollouts, each with 500 steps. We iterate over the data for 100 epochs before proceeding to next iteration of data collection. At each iteration, the size of training data is capped at 50000 steps, randomly sampled from all available trajectories collected. Here we use batch size of 128 and a learning rate of $1e-3$. All models are trained for 150 iterations till convergence. In addition, early stopping is applied if an rolling average of total return decreases.

We apply all the methods with MPC for action selection and control, and report in Table 3 the average return computed over 500 episodes. Again, red numbers denote the best performance and the black ones represent the oracle performance. In all tasks, HyperDynamics is able to infer accurate system properties and generate corresponding dynamics models that match the oracle *Expert-Ens* on seen terrains, and shows a great advantage over it when tested on unseen terrains. *Recurrent* also performs reasonably well on *Ant-Slope*, but our model outperforms both *MB-MAML* and *Recurrent* on most of the tasks. Note that here *Recurrent* can be viewed as serving the same role of the *Direct* baseline in pushing, since it feeds system features together with the states to a fixed global dynamics model. The results suggest that our explicit factorization of system features and the multiplicative way of aggregating it with low-dimensional states provide a clear advantage over methods that either train a fixed global model or perform online parameter adaptation on a meta-trained dynamics prior.

## 5 CONCLUSION

We presented HyperDynamics, a dynamics meta-learning framework that conditions on system features inferred from observations and interactions with the environment to generate parameters for dynamics models dedicated to the observed system on the fly. We evaluate our framework in the context of object pushing and locomotion. Our experimental evaluations show that dynamics models generated by HyperDynamics perform on par with an ensemble of directly trained experts in the training environments, while other baselines fail to do so. In addition, our framework is able to transfer knowledge acquired from seen systems to novel systems with unseen properties, even when only 24 distinct objects are used for training in the object pushing task. Our method presents explicit factorizations of system properties and state representations, and provides a multiplicative way for them to interact, resulting in a performance gain over both methods that employ a global yet fixed dynamics models, and methods using gradient-based meta-optimization. The proposed framework is agnostic to the specific form of dynamics learning tasks, and we hope this work could stimulate future research into weight generation for adaptation of dynamics models in more versatile tasks and environments. Handling more varied visual tasks, predicting both the architecture and the parameters of the target dynamics model, and applying our method in real-world scenarios are interesting avenues for future work. For pushing, we believe our method has the potential to transfer to real-world without heavy fine-tuning, since it uses a geometry-aware representation, as suggested in Tung et al. (2020). For locomotion, our model should also be easily trainable with data collected in real-world, following the pipeline described in Nagabandi et al. (2019).

ACKNOWLEDGMENTS

This paper is based upon work supported by Sony AI, ARL W911NF1820218, and DARPA Common Sense program. Fish Tung is supported by Yahoo InMind Fellowship and Siemens Future-Maker Fellowship.

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

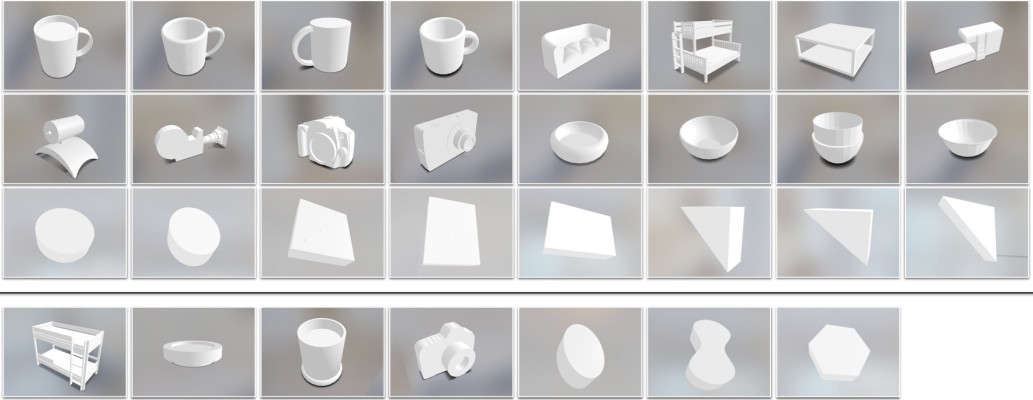

**Figure 3: Top**: 24 objects used for training. **Bottom**: 7 objects used for testing. Objects selected from the ShapeNet Dataset (Chang et al., 2015) are realistic objects seen during daily life, and objects from the MIT Push dataset (Yu et al., 2016) consist of basic geometries such as rectangle, triangle, and ellipse. Note that the shape of the testing objects differ from those used for training. (The third *mug* object used for testing is handleless.)

## A   EXPERIMENTAL DETAILS

### A.1   OBJECT PUSHING

Our dataset consists of 31 object meshes with distinct shapes, including 11 objects from the MIT Push dataset (Yu et al., 2016) and 20 objects selected from four categories (*camera*, *mug*, *bowl* and *bed*) in the ShapeNet Dataset (Chang et al., 2015). 24 objects are used for generating the training data and 7 objects are used for testing (see Figure 3). The size of the objects ranges from $7cm$ to $12cm$. For each pushing trajectory, we randomly sample an object with randomized mass and friction coefficient, and place it on a $0.6m \times 0.6m$ table surface at a uniformly sampled random position. At the begining of each trajectory, we instantiate the robot's end-effector nearby the object, with a random distance from the object's center ranging from $6cm$ to $10cm$. When sampling pushing actions, with a probability of $0.2$ the agent pushes towards a completely random direction, and with a probability of $0.8$ it pushes towards a point randomly sampled in the object's bounding box. Each action is a displacement with a magnitude sampled from $[3cm, 6cm]$.

Our method is viewpoint-invariant and relies on a single 2.5D image as input. We place cameras at 9 different views uniformly with different azimuth angles ranging from the left side to the right side of the robot, and the same elevation angles of 60 degrees. All cameras are looking at $0.1m$ above the center of the table, and $1m$ away from the look-at point. At test time, our model is tested with a single and randomly selected view. All images are $128 \times 128$.

For object pushing with MPC, we place a randomly selected object in the scene with a random initial position, and sample a random target location for it. At each step, we evaluate 30 random action sequences with the model before taking an action. For pushing without obstacle, the maximum distance of the target to the initial position for each object is capped at $0.25m$, and a planning horizon of 1 step is used since greedy action selection suffices for it. For pushing with obstacles, we randomly place 2 additional obstacles in the scene, selected from the same pool of available objects. We place the first obstacle between the starting and goal positions with a small perturbation sampled from $[-2cm, 2cm]$, so that there exists no straight collision-free path from the starting position to the goal. The second obstacle is placed randomly with a distance from the first one ranging in $[15cm, 25cm]$. The distance from the target to the initial position is uniformly sampled from the range $[0.24m, 0.40m]$. The model needs to plan a collision free path and thus a planning horizon of 10 steps is used. For both tasks, we run 50 trials and evaluate the success rate, where it is considered successful if the object ends up within $4cm$ from the target position, without colliding with any obstacles along the way.

**Additional model details.** HyperDynamics for pushing follows the general formulation discussed in Section 3. It's composed of a visual encoder $E_{vis}$, an interaction encoder $E_{int}$ and a hypernet-

work $\mathbf{H}$. These components are trained together to minimize the dynamics prediction loss $\mathcal{L}_{pred}$, as defined in Equation 1. Meanwhile, $E_{vis}$ is also trained jointly for object detection and shape reconstruction. Our $E_{vis}$ follows the architecture of GRNNs (Tung et al., 2019), which uses a similar detection architecture as Mask R-CNN (He et al., 2017), except that it takes 2.5D input and produces 3D bounding boxes. The detection loss $\mathcal{L}_{det}$ is identical as the one defined in He et al. (2017), supervised using ground-truth object positions provided by the simulator. $E_{vis}$ then uses the bounding boxes to crop out a fixed-size feature map for the object under pushing, and encodes it into a visual code $z_{vis}$. In order to help $E_{vis}$ produce an informative $z_{vis}$, we added an auxiliary shape reconstruction loss, where $z_{vis}$ is passed into a decoder $D_{vis}$ to reconstruct the object shape $V^{obj}$. In practice, since our tasks concern planar pushing, we found reconstructing a top-down 2D shape instead of the full 3D one suffices to produce informative $z_{vis}$. This is essentially performing view prediction for the top-down viewpoint. The network is trained to regress to the ground-truth top-down view of the object, which is provided by the simualtor and does not have any occlusion. The view prediction loss $\mathcal{L}_{vp}$ of GRNNs uses a standard cross-entropy pixel matching loss, where the pixel intensity is normalized. We refer readers to (Tung et al., 2019) for more details. The final loss for HyperDynamics for pushing reads:

$$\mathcal{L} = \mathcal{L}_{pred} + \mathcal{L}_{det} + \mathcal{L}_{vp} \tag{2}$$

Note that while it's expensive to collect actual pushing data with objects, data consisting of various shapes are easily accessible. In order to ensure $z_{vis}$ produced by the shape encoder $E_{shape}$ captures sufficient shape information, we train it on the whole ShapeNet (Chang et al., 2015) for shape autoencoding (essentially top-down view prediction) to jointly optimize for $\mathcal{L}_{vp}$. In practice, we load random objects in the simulation and collect rendered RGB-D images from randomized viewpoints. During training, data fed into $E_{vis}$ alternates between such rendered data and images of actual objects being pushed, while the other components are only trained with the objects being pushed. Note that such auxiliary data is only used to optimize for $\mathcal{L}_{vp}$. (It's also possible to use such data to optimize for $\mathcal{L}_{det}$, but in our experiments training object detection with only the objects being pushed suffices.) The reason for doing this is that we can collect such visual data very fast using easily accessible object meshes, while we only need to interact with a few objects with actual pushing. This helps our model generalize to novel objects unseen during actual pushing, since $E_{vis}$ is now able to produce a meaningful $z_{vis}$ for these novel objects.

## A.2    Locomotion

The *Slope* environment consists of a series of consecutive upward slopes with length of $15m$. For training, the height of each slope is randomly sampled from $[0.5m, 3.5m]$, and for testing, the height is randomly sampled from $[0m, 0.25m] \cup [3.75m, 4m]$. The *Pier* environment consists of a series of consecutive blocks with length of $4m$. For training, the damping coefficient of each block is randomly sampled from $[0.2, 0.8]$, and for testing, the damping coefficient is randomly sampled from $[0, 0.1] \cup [0.9, 1.0]$. 5 experts are trained to form the expert ensemble for each task. The *half-cheetah* robot has 6 controllable joints while the *ant* has 10. The reward function for *half-cheetah* is defined to be $\frac{x_{t+1} - x_t}{0.01} - 0.05\|a_t\|_2^2$, and for *ant* it is $\frac{x_{t+1} - x_t}{0.02} + 0.05$, where $x_t$ refers to the x-coordinate of the agent at time $t$.

## B    Additional Experimental Results

### B.1    Model Ablations

In order to evaluate the effect of each network component in our model and motivate our design choices, we present an ablation study here, and report the performance of each ablated model in Table 4 and 5. For pushing, our model contains a visual decoder for top-down shape reconstruction (top-down view prediction), uses a cropping step to produce object-centric feature maps, and uses $z_{vis} \in \mathbb{R}^8$ and $z_{int} \in \mathbb{R}^2$. For locomotion, our model uses $z_{int} \in \mathbb{R}^2$. We vary the sizes of these latent codes and evaluate how they affect the model performance. In addition, in order to help understand the performance of the 3D detector in our visual encoder, we also report the prediction error using ground-truth object positions as input. The results in the table indicates that for pushing, it's essential to include the visual decoder to regularize the training, otherwise the model overfits badly on the training data. The cropping step is also important and the object-centric feature map

**Table 4:** Prediction error of HyperDynamics for pushing with different model ablations.

| Model | Ours | w/ gt object positions | w/o visual decoder | w/o cropping | $z_{vis} \in \mathbb{R}^{16}$ | $z_{vis} \in \mathbb{R}^{32}$ | $z_{int} \in \mathbb{R}^{4}$ | $z_{int} \in \mathbb{R}^{8}$ |
|---|---|---|---|---|---|---|---|---|
| Error (seen) | **0.83±0.35** | **0.76±0.32** | 0.87±0.44 | 1.45±0.64 | **0.82±0.43** | **0.84±0.37** | **0.83±0.39** | **0.81±0.34** |
| Error (novel) | **0.93±0.53** | **0.86±0.44** | 4.32±2.08 | 1.76±0.90 | **0.94±0.51** | **0.96±0.49** | **0.93±0.52** | **0.94±0.55** |

**Table 5:** Average total return of HyperDynamics for locomotion with different model ablations.

| Model | Cheetah-Slope | | Cheetah-Pier | | Ant-Slope | | Ant-Pier | |
|---|---|---|---|---|---|---|---|---|
| | Seen | Novel | Seen | Novel | Seen | Novel | Seen | Novel |
| Ours | 107.9±63.1 | 124.5±64.7 | 349.4±189.0 | 337.2±179.4 | 138.8±65.3 | 140.1±72.0 | 216.6±121.2 | 208.4±103.8 |
| $z_{int} \in \mathbb{R}^{4}$ | 103.5±59.0 | 127.6±62.9 | 351.2±176.4 | 333.5±167.9 | 141.2±69.3 | 137.1±73.2 | 221.2±115.9 | 202.1±93.0 |
| $z_{int} \in \mathbb{R}^{8}$ | 112.1±64.0 | 122.9±59.9 | 363.7±173.6 | 329.8±173.1 | 134.5±63.8 | 134.9±75.8 | 214.2±124.1 | 206.7±97.3 |

**Table 6:** Comparison of prediction error ($\times 10^{-2}$) for locomotion tasks.

| Model | Cheetah-Slope | | Cheetah-Pier | | Ant-Slope | | Ant-Pier | |
|---|---|---|---|---|---|---|---|---|
| | Seen | Novel | Seen | Novel | Seen | Novel | Seen | Novel |
| Expert-Ens | **3.4±0.6** | 26.1±6.4 | **5.4±0.8** | 12.0±6.1 | **8.9±3.6** | 25.4±9.2 | **14.2±4.7** | 41.7±13.2 |
| Recurrent | 5.2±1.7 | 22.4±5.2 | 5.2±1.2 | 12.1±4.5 | **9.1±3.2** | **23.0±8.9** | 18.3±8.0 | 39.1±18.0 |
| MB-MAML | 5.7±2.2 | 21.0±9.2 | 5.1±1.3 | 12.4±6.7 | 12.8±5.2 | 29.1±11.2 | 18.9±7.2 | 42.3±19.7 |
| HyperDynamics | **3.5±0.6** | **17.9±7.3** | **5.6±1.0** | **11.3±4.7** | **9.3±3.4** | **22.8±7.9** | **14.7±5.3** | **33.2±14.2** |

helps the model to make more accurate predictions. As for the object detector, when compared to the model using ground-truth object positions, our model using detected object positions only shows a minor performance drop, suggesting that the 3D detector in the visual encoder is able to accurately detect the object locations. The table also motivates our choice for the dimension of the latent code $z_{vis}$ (pushing) and $z_{int}$ (both pushing and locomotion), as further increasing their sizes does not result in any clear performance gain. (In Table 5, none of the numbers is marked in red as the results are similar across all ablated models.)

## B.2 ADDITIONAL RESULTS ON PREDICTION ERROR FOR LOCOMOTION

In the locomotion tasks, the data collected is not i.i.d., since it depends on the model that's being optimized online. In order to show a clear comparison between our method and baselines regarding their prediction accuracy, we train HyperDynamics till convergence for each task, and use this converged model with MPC to collected a fixed dataset. Then, we compare the prediction error of all models on this single dataset and report the results in Table 6. The error is computed as the mean squared error between the predicted state and the actual state in the robot's state space. Again, our model is able to match the performance of *Expert-Ens* on the seen terrains and still performs reasonably well on unseen terrains, outperforming the baselines consistently.

