# OpenReview forum: "HyperDynamics: Meta-Learning Object and Agent Dynamics with Hypernetworks"
_ICLR.cc/2021/Conference — ICLR 2021 Poster_

### Official Review · AnonReviewer1 · 2020-10-27
**Great idea, slightly flawed paper but probably possible to improve in rebuttal period.**

**Rating:** 7
**Confidence:** 4

**Review:**

## Summary

The authors present "HyperDynamics", a novel method for system-identification and learning of flexible forward models that can be used in planning tasks. The presented methods is generic and is shown on both locomotion and pushing tasks with different simulated robots.

I enjoyed reading this work a lot and I hope it gets accepted. It's a clever idea and most flaws that I'm about to point out are easily addressable by the authors.

## Strengths & Weaknesses

#### Strengths

1) The method is generic (shown to work across tasks and environments).
2) The baselines are strong. When I started reading your paper, I thought that DensePhysNet and some form of MAML would be good candidates for this to compare again and it turns out these were indeed included.
3) Figure 1 + caption as well as the introduction to section 3 do a great job at introducing the architecture in a way that would allow the reader to create a basic implementation.
4) Code was included. I didn't run it but it's clean and seems functional from what I can tell.

#### Weaknesses

1) You really, really need to be more clear in the main paper on the implementation details. You can't move the amount of training data to the appendix and it's not good practice to only include the network architecture by name in the main paper. And what are all your losses? You make the method looks super simple but then you train on shapenet, some 2D reconstruction, something about cropping, and there's a GRU in there too (full backprop vs truncated backprop?). The appendix shines a _little_ light on this but you need to be way more specific in the main paper. That has to stand on its own.
2) You don't motivate all the nitty-gritty implementation choices. Why did you add the decoder? What's the performance if you remove it? What about the cropping? What if you don't do an object-centric feature map, but instead a few CNN layers? What are the individual contributions of all these details?
3) DensePhysNet, Visual Foresight, and many other works in this domain use (simple) real-robot experiments to demonstrate that their method can handle realistic robot noise. Obviously there's a global pandemic happening at the moment, so I won't require you adding this for the rebuttal, but I think in order to really establish this method (maybe before putting it on Arxiv), you'd have to add some real-robot experiments. This can be as simple as a 180USD RealSense and a 500USD robot arm plus a few objects and a playfield. It's become a standard for system-identification-style works and it's justified in my opinion since your method isn't inherently useful in simulation where the user has access to all the information and can arbitrarily reset/reposition the model. And since you don't have ShapeNet data for many real-world objects (which you seem to need for pretraining), could you at least add a sentence or two detailing how this would transfer to real-world problems?

**TL;DR my main requests:** (2) Motivate implementation details (add ablations if you have any) and (1) be more explicit about them in the main part of the paper.

## Impact & Recommendation

Despite that there seem to be a lot of hacks that make this method in the specific settings, I think the general idea behind it is sound. And I think the authors show that it performs better than the sota, at least in simulation. Therefore I'd recommend acceptance given that the authors add the requested information. In its current shape, it's a 6 for me but if my main concerns are addressed, I'm happy to up this to a 7 or if major improvements are made and my questions below are answered, to an 8.

## Questions, Nitpicks, Comments

- Kudos for not making another acronym method
- There are a lot of typos and orthographic errors, would recommend a spell-checked or getting this proofed. Examples: section 2 "poinclouds", section 2 "properties in hand" -> "properties at hand"
- Maybe start the introduction with an example, e.g. how children are able to chew on a block of wood to assess its hardness and then build towers with it.
- **important** Introduction: when you go over (i-iv), that feels a bit too long and lit-reviewy and misplaced in the introduction. I would recommend the following changes: (a) trim this severely, only mention that there are model-based methods that usually do only one environment and there's meta-learning and how your method is more adaptable than either, (b) move this into the literature section, where you have to come back to it anyway, (c) move the Hypernetworks section from the literature into a separate "Background" section and develop it a bit further, since it's less "competing method" and more "you should know about this to understand out method".
- Also in the introduction, you present (i-iv), and you mention how your method is better/different than (i-iii) but you never address (iv).
- It's become a standard to summarize the contributions again at the end of the introduction, ideally as bullet points. Please add these.
- In equation (1), why is the ordering O-T-N for the sums? I feel like ONT would be more natural, no?
- When reading the method, my main question was if the method would work on "dense" trajectories or on before-and-after photos like DensePhysNet. This is only answered a few pages later but I think this belongs in 3-Overview or 3.1. Just to be clear, you're gathering trajectories of length 4s, i.e. 5 frames of 800ms where you do NOT retract the robot arm when pushing, right? (Compared to DensePhysNet, where the arm is never visible because they take photos before and after complete standstill). If that's the case, how do you deal with occlusion from the arm?
- Do you encourage object-object interactions in any way or do they just occur randomly? Or do you only ever experiment with single objects?
- The object orientation vs state section isn't super clear? You're subtracting an object's absolute starting position+orientation from it's future trajectory points?
- In 3.2: Why a GRU, why not LSTM? Why k=16 (and similarly why k=5)... This ties into the main criticism from above. Please motivate your choices.
- In 4.1: I think this is a typo, but it says you added beds to your experiment table. I think they'd be a bit too large, no? :D
- 4.1: specify the random mass+friction range, please!
- 4.1: same with the total amount of training data/frames
- And since you won't have ShapeNet
- 4.2: I think it's a half-cheetah, not a cheetah.
- 4.2: I don't understand why it's unrealistic to assume arbitrary resetting in simulation. That's one of the benefits of running simulations and common practice.
- 5: What do you mean "predicting both the structure and parameters of the target dynamics model"? Parameters is clear (mass, friction, etc.) but what's the structure here?

---

> ### Comment · AnonReviewer1 · 2020-11-24
> **Slightly disappointed**
>
> I have to say, I'm slightly disappointed that the authors didn't respond to the reviewers within the rebuttal period.
> There was probably a reason for this, so I would appreciate it if the authors could at least mention what happened or why they chose not to respond.

---

> > ### Author Response · Authors · 2020-11-25
> > **Sorry**
> >
> > We are so so sorry for keeping you waiting! We believe the deadline for rebuttal hasn't passed yet (it's due 24th anywhere on Earth) and we have now updated our manuscript together with detailed rebuttals. We apologize for the last minute update.  It took much longer than we expected to finish as we added many more experiments and visualizations in the updated version and we wanted to address all concerns in the reviews carefully. We hope the updated version addresses all of your concerns!

---

> ### Author Response · Authors · 2020-11-25
> **Response to review (part 1/4)**
>
> We are really encouraged that you liked our idea and agreed with our baseline choices, and we really appreciate your comments and suggestions with so much details! We hope to address all of your concerns below.
>
> (1) Lack of implementation details in main paper.
>
> Thanks for raising this issue. We added more detailed description into the ‘baseline’ paragraphs, and also added a paragraph (Implementation Details) after the baseline paragraphs in both Section 4.1 and 4.2. We have moved the architecture details of our model from the appendix to there. We also added more details of the baselines (architecture, training, etc.) and the amount of training data in these sections. To briefly summarize here, MB-MAML takes the exact same input as our model does, except that it uses interaction data for online model adaptation. The Direct and Recurrent  baselines use the same input and use the same visual and interaction encoders  as our model does in their corresponding tasks. All the dynamics model used in Direct, Recurrent, XYZ, MB-MAML and Expert-Ens use the same architecture. VF and DensePhysNet are implemented using the architectures described in their corresponding papers. The GRU is trained with full backpropagation and we have added this to the ‘Implementation Details’ section too.
>
> Regarding the training of the visual encoder (object detection, view prediction, losses, etc.), we have added a detailed description in Appendix A.1. Since it’s a bit lengthy and it’s built mostly based on GRNNs, which is not the main contribution of this work, we think it’s reasonable to put it in Appendix, and referred to it in relevant places in the main paper.
>
> (2) Details of implementation choices
>
> Thanks for bringing this up! The decoder is crucial since it helps regularize the training of the visual encoder and makes sure the produced z_vis captures sufficient shape information. The cropping step is also important to produce the object-centric feature map, and drives the model’s attention to the object under pushing. In order to help readers better understand these components, we have added an ablation study in Appendix B.1. We hope the results there could explain our implementation choices.

---

> > ### Author Response · Authors · 2020-11-25
> > **Response to review (part 2/4)**
> >
> > (3) Transfer to real world.
> >
> > Thanks for suggesting this! Yes, testing our method in real world is definitely our next step. Real-world application is indeed a very important and many methods have difficulty in transferring directly to real world scenario mainly because the data distribution seen in real world differ significantly from the simulated data, especially the visual inputs. Our method is shown to generalize better to unseen environment in simulation. As for real world scenarios, a very recent paper [5] also proposes to use GRNNs as the perception frontend to learn 3D object motion prediction under pushing, which enables viewpoint invariance of the learned dynamics model and shows good performance in real world when applying their model trained solely in simulation. (Their main contribution is to build a graph network over object-centric feature maps, while we focus on leveraging hypernetworks for generalization). The authors claim that their model’s good sim-to-real transferrability is achieved due to the fact that geometry information is shared by simulation and the real world by a large extent, and their models uses a 3D geometry-aware representation produced by GRNNs. Hyperdynamics for pushing also possesses such properties as it uses the same geometry-aware visual encoder, so we expect it to produce reasonable performance when transferred to real world. Also, indeed we need to train the visual encoder on ShapeNet as an auxiliary task, but we believe sim-to-real transfer of such shape encoding capability should also be possible due to the same reason, as it’s powered by the geometry-aware visual encoder. If direct sim-to-real doesn’t work satisfactorily, it’s still possible to collect such data in a self-supervised way to finetune the model, since as shown in our experiments, we don’t need to collect additional expensive data of actual object pushing, and we only need to collect visual data by asking a robot to move an attached camera around. As for locomotion, our experiments follow the same setup as the MB-MAML [4] baseline, and uses the exact same input data. The locomotion task setting does not rely on visual input, so the mismatch between simulation and the real world should be less significant compared to the pushing setting.  In [4] the authors trained their method in real-world and claimed good performance. In our experimental section we do show that HyperDynamics for locomotion performs better than MB-MAML to unseen environments in simulation, and we see no major bottlenecks for collecting data and training our method in the real world following the same way. Therefore, we expect it to also produce a good (if not better) performance in a real world locomotion setting.
> >
> > We have added a few sentences in the conclusion section to list real world applications as our future work, and briefly talked about how we expect it to work.
> >
> > ### Questions
> > - Typos and errors
> >
> >  We apologize for having those errors in the original paper. We have proofread the paper and corrected the errors.
> >
> > - Start the introduction with an example
> >
> >  We do have an example in the first paragraph about “moving an object on the ground”. Maybe it’s not as interesting as the children example you suggested :) but we hope it illustrates clearly the point that human beings are able to infer object properties via observation/interactions.
> >
> >  - Restructure (i-iv) and literature review
> >
> >  Thanks for suggesting this! We have followed your advice and shortened the groupings of current literature, and moved the groupings in to the related work section (section 2.1). We have also extended the "hypernetwork" paragraph with more explanations and more references to related works. We feel it might be not sufficient to stand on it own as a separate section, so we put it in an independent subsection now (section 2.2) and named the title "background on hypernetworks". We hope the structure of the paper is more clear now!
> >
> > - Address (iv) in introduction.
> >
> >  We believe this is addressed now. In the new introduction we mentioned it and stated that we outperformed it, and in section 2.1 again we stated how our method differs from the 4 types of methods in the literature.
> >
> > - Summarize the contributions again at the end of the introduction
> >
> >  Thanks for suggesting this! We have summarized our contributions with bullet points at the end of the introduction.
> >
> > - OTN vs ONT
> >
> >  Yes we agree. We have changed the order to ONT now.

---

> > > ### Author Response · Authors · 2020-11-25
> > > **Response to review (part 3/4)**
> > >
> > > - Retract arm?
> > >
> > > 	No, we do not retract arm when collecting the pushing data. The Kuka robot we use in simulation has a single rod-shaped end-effector (as shown in Figure 1). The end-effector is not very bulky so it only introduces minor occlusions. The GRNNs we use learns to infer shape and handle such minor occlusions implicitly (it handles single RGB-D input and learns to complete missing shape information via self-supervised top-down view prediction, as demonstrated in  [1]). We have added a sentence to mention this explicitly in the 4th paragraph in section 3.1. Also, we hope the newly added details of E_vis in Appendix A.1 and the qualitative results of view prediction in Appendix B.4 (Figure 5) would help clarify this.
> > >
> > > - Object-object interaction
> > >
> > >  No, in the data we collected there’s only robot-object interaction. This is because the latent code z is object-centric and we consider the properties of the specific object under pushing as the contextual information of the dynamical system. The main purpose of this paper is not to present a system dedicated to modeling object interactions, rather, we are trying to show introducing hypernetworks helps handle system properties and states in a more structured way, and helps the model to generalize better. That said, extending current framework to handle object-object interactions is definitely a very interesting future direction. One possible way is instead of conditioning on the properties of a single object, we construct a graph-structured encoder using a graph network that operates over object nodes, encode their properties into a scene-level embedding, and pass it into the hypernetwork, which can then generate a dedicated dynamics model for each of the objects in the scene.
> > >
> > > - Object orientation vs state
> > >  Thanks for pointing out this! We apologize for causing the confusion about our orientation representation. To make it more clear, consider an object oriented at orientations A and B; we now have 2 design choices:
> > >    - (1) treating them as the same object but with different orientation state, then we would feed its canonical shape into E_vis, and feed different orientations into the generated dynamics model (since the dynamics model takes as input the object state), or
> > >    - (2) simply treating them as 2 different objects as if they have different shapes! (Imagine you consider a rotated mug as a new mug.) Then we can simply remove the orientation info from its state, only feed its position info (in addition to other system states) into the generated dynamics model, and encode their shape (which already includes their orientation) into the shape code z_vis.
> > >
> > >  If we choose (1), then the problem is that when trained on only 24 objects, the model would only see 24 distinct z_vis, which could result in very bad overfitting for both E_vis and Hypernetwork H. So in our framework we chose (2), which enables both E_vis and H to see a much more smooth distribution of input data, and this helps our method to generalize. In practice, we did try (1) at first but the model failed to generalize.
> > >
> > >  For unrolling, at any timestep t, we define the current orientation of object as (0,0,0,1) (quaternion without any rotation), and compose it with the predicted delta quaternion to obtain its quaternion for next timestep t+1. After that, we reorient the object feature map (obtained at timestep t) using this predicted quaternion and treat it as a new object to feed into E_vis, which produces z_vis and feeds into H. Therefore, this updated orientation is encoded in the weights of the generated dynamics model, and not fed as input to it.
> > >
> > >  We are not doing the same thing with object positions. The absolute position is fed into the generated dynamics model as part of the state input. We hope this clarifies your doubt. We have also updated the last paragraph in Section 3.1 to make this more clear.

---

> > > > ### Author Response · Authors · 2020-11-25
> > > > **Response to review (part 4/4)**
> > > >
> > > >  - GRU vs LSTM; value of k
> > > >
> > > >  We adopted GRU following the Recurrent baseline, as our model and Recurrent uses the same recurrent encoder, in order to have a fair comparison. In this work, we are not striving to investigate which is the best recurrent encoder architecture for locomotion tasks; instead, we are trying to show the advantage of leveraging hypernetworks in dynamics learning. So, we aim for simplicity of the encoder modules, and we believe the comparisons we presented is reasonable and clear. We believe now the updated experimental section would make this clear, since we added details of the baselines. In practice, we did try with LSTMs and it did not give us any performance gain. Similarly, the k values we used are found to be just sufficient to encode the interaction data. Using a bigger value than these does not help improve performance. We have added two short sentences to where the two number are introduced, to explicitly mention this. Also, we included some visualizations of the learned latent space for z_int in Appendix B.4, and compared between different values of K. We hope these could further motivate our choices.
> > > >
> > > >  - Bed in ShapeNet
> > > >
> > > >  We do have bed in our dataset. The objects in ShapeNet all have normalized sizes and we downscaled all of them by a fixed constant. The objects used in our data collection have sizes ranging from 7cm to 12cm, as mentioned in section A.1. We have included Figure 3 in appendix to show the all objects used.
> > > >
> > > > - More details in section 4.1
> > > >
> > > >  We have moved the mass+friction range from appendix to section 4.1. We have also added the amount of training data into the "implementation details" paragraph in 4.1.
> > > >
> > > > - Half cheetah
> > > >
> > > >  We have now corrected this when talking about the robots in both the main paper and the appendix. We kept task names unchanged (Cheetah-Slope and Cheetah-Pier) to make them simple and consistent with the other 2 tasks.
> > > >
> > > > - Resetting in simulation
> > > >
> > > >  We apologize for the misleading sentence. We actually meant that for pushing tasks, it’s reasonable to reset the object positions and orientation with uniformly sample state in its state space. While in locomotion tasks, randomly sampled states in the agent’s joint space do not match the true distribution that are seen during actual locomotion. We have updated the sentence to explain this.
> > > >
> > > > - Predicting both the structure and parameters of the target dynamics model.
> > > >
> > > >  By "parameters" we meant the weights of the generated dynamics model, and by "structure" we meant the architecture of the dynamics model. There are some interesting works that use hypernetworks for neural architecture search [2][3] in the literature, and we think it would be promising to bring that capability into dynamics learning to find the best architecture for the forward dynamics model. We have changed the word ‘structure’ to ‘architecture’ and hope it’s more clear now.
> > > >
> > > >
> > > > We would like to thank you again for such a detailed feedback! We hope the revised manuscript now presents and explains our method better.
> > > >
> > > > [1] Hsiao-Yu Fish Tung, Ricson Cheng, and Katerina Fragkiadaki. Learning spatial common sense with geometry-aware recurrent networks. In Proceedings of the IEEE Conference on Computer Vision and Pattern Recognition, pp. 2595–2603, 2019.
> > > >
> > > > [2] Chris  Zhang,  Mengye  Ren,  and  Raquel  Urtasun.    Graph  hypernetworks  for  neural  architecture search. In International Conference on Learning Representations, 2019.
> > > >
> > > > [3] Andrew Brock, Theodore Lim, James M Ritchie, and Nick Weston.  Smash: one-shot model archi-tecture search through hypernetworks. In International Conference on Learning Representations, 2018.
> > > >
> > > > [4] Anusha Nagabandi, Ignasi Clavera, Simin Liu, Ronald S. Fearing, Pieter Abbeel, Sergey Levine, and Chelsea Finn. Learning to adapt in dynamic, real-world environments through metareinforcement learning. In International Conference on Learning Representations, 2019.
> > > >
> > > > [5] Hsiao-Yu Fish Tung, Zhou Xian, Mihir Prabhudesai, Shamit Lal, Katerina Fragkiadaki. 3D-OES: Viewpoint-Invariant Object-Factorized Environment Simulators. the Conference on Robot Learning (CoRL), 2020.

---

### Official Review · AnonReviewer2 · 2020-10-29
**An interesting idea held back by unconvincing experimental design**

**Rating:** 6
**Confidence:** 3

**Review:**

== Update ==

Thank you for your detailed response. The newly added clarifications and sanity checks have greatly improved the quality of the paper, and I am therefore increasing my score from 4 to 6. I believe the model capacity comparison (Table 6) is especially important for demonstrating the value of the new architecture, and would recommend mentioning that result in the main paper.

== Original Review ==

The paper proposes a model for predicting the dynamics of a physical system based on hypernetworks:
given some observed interactions and some visual input, the hypernetwork outputs the parameters of a
dynamics model, which then predicts the evolution of the system's state over time. Experiments are
conducted on an object pushing and a locomotion task.


Strengths:
 1. The paper addresses an important question, namely, how a dynamics model may adapt to
    environments that don't fully match its training distribution.
 2. The proposed use of a hypernetwork is plausible and novel to my knowledge.
 3. The related work section appears comprehensive, and, to my knowledge,  does not miss any major
    prior work.

Weaknesses:
 1. The main claim of the paper is that hyperdynamics network offers better prediction accuracy and
    generalization than a standard dynamics model. I feel like the evaluation of this question is
    confounded by the choice of tasks and baselines. On the pushing benchmark, the XYZ, VF, and
    DensePhysNet operate on different modalities than HyperDynamics (either no state information or
    no visual information), and are therefore difficult to compare. For the MB-MAML baseline, this
    is not specified. The Expert-Ens model cannot be expected to generalize, since it is designed to
    overfit on individual objects. As a result, only the 'Direct' baseline clearly operates in the
    same experimental regime as HyperDynamics. However, nothing is reported on the model
    architecture or the training method for that baseline, raising the question if its model
    capacity was competitive. My impression is that this experimental design blurs the effects of
    (a) using side-information to infer system properties, and (b) utilizing such information
    through a hypernetwork as opposed to a standard dynamics predictor. If the goal is to evaluate
    the new architecture, these should be disentangled.
 2. On the locomotion benchmark, the Recurrent baseline is similarly unclear.  Sanchez-Gonzalez et
    al. is cited, but that paper focusses on comparing recurrent models based on graph networks to
    those based on MLPs, and it is unclear which model was used.
 3. No results are reported for the prediction accuracy on the locomotion task, which would have
    helped evaluate the performance of the dynamics models more directly than the task scores.
 4. Many of these issues could have been avoided by testing on established benchmarks from the
    literature, for which results are available. If there is a simulator available, generalization
    ability could still have been tested by varying the physical constants of the dataset.
 5. The paper contains a decent amount of typos and grammatical errors.


Overall, while the paper presents an interesting idea, the experimental evaluation is not convincing
in its current state: Baseline architectures are not fully specified, many of them did not receive
the same input, and no benchmark task with previously reported results has been used. As a result, I
recommend rejection at this time.


Questions:
 1. In eq. 1, should omega be a parameter of H(.) instead of F(.)?
 2. Section 3.1 introduces the "1-dimensional code $z_{int} \in \mathbb{R}^2$". So is it one or
    two-dimensional?
 3. Overall, the dimensionality of the latent codes and hidden layers seems incredibly small, e.g.,
    only 1/2 numbers to encode prior interactions, and 8 to encode shape. Is there really no benefit
    to using higher capacity models?

---

> ### Author Response · Authors · 2020-11-25
> **Response to review (part 1/2)**
>
> Thank you for the detailed comments and criticisms! We feel very encouraged that you found the question we study important and our approach novel. We hope to address all your concerns below.
>
> (1) Design and details of the baselines.
>
> We apologize for missing important details of the baselines and we really appreciate you for raising this issue. Indeed, your description of the purpose of each baseline matches closely to our intention, and we would like to explain them in further details here:
> - The XYZ baseline is more like an ablation where we want to evaluate how a standard dynamics model performs without having access to the visual features and physics properties.
> - You are absolutely right that VF and DensePhysNet operate on different modalities than our method. We included them in the paper because we believe it would be good to compare our method with some image-based methods in the literature. We think this makes the comparison more complete.
> - MB-MAML uses the exact same architecture as the generated dynamics model of HyperDynamics, it also uses the exact same input data. The only difference is that it updates the model parameters online using the interaction data.
> - Expert-Ens is included because we want to evaluate if our model generate good expert online that can match the performance of these separately trained ones. Each expert in Expert-Ens uses the exact same architecture as the generated model of HyperDynamics.
> - Direct: you are right that it operates in the same experimental regime as ours. It uses the exact same input as ours (both side information and system state). It uses the same visual encoder and interaction encoder as ours, and feed the concatenated code z into a dynamics model, which follows the exact same architecture as the generated dynamics model of HyperDynamics. We believe this is a clear comparison since it clearly shows adding a hypernetwork to process such system properties could help improve performance.
>
> Again, we are sorry for not explaining these details clearly in the main paper. We have added more details into the baseline description section in the paper. We also added an ‘Implementation Details’ paragraph after the baseline paragraph. It contains the architecture details that were in the appendix, and we also added some more details there to describe the architecture of the baselines.
>
> We also appreciate that you brought up the issue regarding model capacity. Indeed, although the dynamics model in Direct uses the same architecture as the the generated one in HyperDynamics, it needs to process more information than our generated model, since it also handles the latent code z, while in HyperDynamics, z is fed into the hypernetwork. We believe a more fair comparison on model capacity should be between the dynamics model in Direct and the hypernetwork in HyperDynamics (the last layer of the hypernetwork outputs the weights of the generated dynamics model). Considering that the hypernetwork is a fully-connected network with a 16-unit hidden layer, effectively it has around 16x more parameters than the Direct baseline. However, in our experiments we found that further increasing the model capacity of Direct does not help further improve its performance, and would result in either overfitting or underfitting. We apologize for not including such comparison in the original paper. We have now added an analysis on model capacity in Appendix B.2. We hope this shows that our architecture is a more effective and structured way to handle system properties and low-dimensional system states.
>
>
> (2) Recurrent baseline in locomotion task
>
> Yes, we agree that the focus of [1] is the proposed graph networks. We chose the recurrent model in their work because it’s a straightforward architecture and is easy to implement. The purpose of our work is not to investigate which recurrent architecture is the best, and HyperDynamics and the Recurrent baseline uses the same recurrent model to encode interaction trajectories. We believe such comparison is reasonable and does illustrate the advantage of bringing hypernetworks into dynamics learning. We apologize for not explaining this well in the original paper. We have now added more details of the baselines in Section 4.2. We hope it’s more informative now.

---

> > ### Author Response · Authors · 2020-11-25
> > **Response to review (part 2/2)**
> >
> > (3) No prediction accuracy in locomotion task.
> >
> > The reason we did not include this is that data collected in the locomotion setting is not i.i.d: a model is trained first with a round of random data collection, and then the model is used with MPC for further data collection. Therefore, the distribution of the data collected by each method are not identical, and direct comparison on prediction accuracy on different data distribution is not very informative. (In pushing we collect randomly sampled data only once and use that data to train the model, while in locomotion it’s not a good practice since randomly sampled robot states do not match the true distribution encountered during the actual robot running.)
> >
> > However, we do think it’s a good idea to show prediction accuracy comparison in addition to task scores, in order to help readers understand our method better. In order to show prediction accuracy, we now train our HyperDynamics till convergence, and use a converged model with MPC to collect robot locomotion data, and evaluate all models on this single dataset. We have added such results in Appendix B.3. We hope this addresses your concerns.
> >
> > (4) Testing on established benchmarks.
> >
> > Yes, we agree that testing the performance of our method on a well established benchmark would be very informative. However, to the best of our knowledge, though there exists a bunch of standard RL benchmarks, there is not yet a standard benchmark that is specifically designed to evaluate how a method generalizes to system variations and provides standard test ranges for varying physical constants. Both the pushing tasks and the locomotion tasks in our paper are commonly studied RL tasks in the current literature, and we use widely adopted physics simulation engines Bullet and Mujoco for easier comparison. In fact, the pushing setting in our paper follows the exact same pushing setting introduced in [2], except that we added variation in the object’s physical constants. Our locomotion tasks follow exactly the ones proposed in [3], and our MB-MAML baseline uses directly their code without any modification. We believe the experiments conducted in our work and the comparisons are reasonable and informative.
> >
> > (5) Typos and grammatical errors.
> >
> > Thanks for pointing this out! We apologize for having those errors in the original paper. We have corrected the errors and updated the paper.
> >
> > ### Questions
> > 1. Yes it’s a typo. We have corrected it now. Thanks for finding this!
> > 2. We used the phrase ‘1-dimensional code’ to suggest that it’s a vector. We agree that this is misleading and have removed this phrase.
> > 3. For pushing, the latent vector z_int only needs to encode the friction and mass of the object under pushing, which are 2 physical constants. For locomotion, z_int only needs to encode the slope or the damping coefficient. Therefore, such 2-dimensional vectors are enough to encode the interactions. For the object shape, we did run a grid search over various sizes of the latent code, and found no clear performance improvement with z_vis bigger than the current 8-dimensional one. We have added an ablation study in Appendix B.1 to compare different sizes of the latent codes. We hope the results motivate our design choices.
> >
> >
> >
> >
> > [1] Alvaro Sanchez-Gonzalez, Nicolas Heess, Jost Tobias Springenberg, Josh Merel, Martin Riedmiller,Raia Hadsell, and Peter Battaglia. Graph networks as learnable physics engines for inference andcontrol. InInternational Conference on Machine Learning, pp. 4470–4479, 2018b.
> >
> > [2] Hsiao-Yu Fish Tung, Zhou Xian, Mihir Prabhudesai, Shamit Lal, Katerina Fragkiadaki. 3D-OES: Viewpoint-Invariant Object-Factorized Environment Simulators. the Conference on Robot Learning (CoRL), 2020.
> >
> > [3] Anusha Nagabandi, Ignasi Clavera, Simin Liu, Ronald S. Fearing, Pieter Abbeel, Sergey Levine,and  Chelsea  Finn.Learning  to  adapt  in  dynamic,  real-world  environments  through  meta-reinforcement learning. InInternational Conference on Learning Representations, 2019.

---

### Official Review · AnonReviewer4 · 2020-11-01
**Review for HyperDynamics: Generating Expert Dynamics Models by Observation**

**Rating:** 6
**Confidence:** 4

**Review:**

#### Summary:
This paper proposes an adaptive dynamics model based on the idea of hypernetworks. It is demonstrated that this approach compares favorably to other ways of adapting dynamics models such as conditioning on a separate feature input and meta learning by gradient-based model updates. The proposed approach is evaluated on Pushing and Locomotion tasks.


#### Pros:
- The proposed approach for conditioning dynamics models on rollouts to model system-specific properties using the hypernetworks idea seems novel and is interesting.
- Paper is clearly written, the provided figures help understanding
- Outperforms state-of-the-art adaptive dynamics modeling approaches [Nagabandi et al., 2019], [Sanchez-Gonzalez et al., 2018b]
- Reasonable baselines are used for comparison, such as fixed model (XYZ), input feature conditioning (Direct), expert ensemble, and state-of-the-art adaptive dynamics models

#### Cons:
- The paper does not explain training details for the architecture sufficiently well. How are the network components trained, especially the visual recognition part for object pushing? What kind of supervision with ground truth is required to train the components, for instance for object detection and shape representation? Are components pretrained and how? Which losses/data are used for training?
- Its unclear why moving from a canonical to an oriented shape representation in Sec. 3.1 should improve results. Shouldnt this limit generalization and require more training data?
- Giving standard deviations in addition to the average values in table 1-3 would complete the numerical results
- Sec. 1) Why is PlaNet [Hafner et al., 2019] listed as "no adaptation", although it contains a recurrent state representation?
- It appears magical that the approach performs better on novel than on seen objects during training for Cheetah-Slope or Ant-Slope in Table 3. Please discuss.

#### Recommendation:
The paper reads well and proposes an interesting novel approach which could deserve acceptance.
The paper should address the points raised in paper weaknesses.

#### Questions for rebuttal:
Please address points raised above in "weaknesses".

#### Typos:
- p2: "They are are"
- p4: "E_int then maps z_int to a 1-dimensional code z_int ∈ R2" - shouldn't this be "E_int maps interactions to 2-dimensional code z_int ∈ R2"?
- p4: "which is typically comprised of an agent and its external environment" -> "which typically comprises / which is typically composed of"
- Table 1: Motion rediction error -> Motion prediction error
- Advice: In Figure 1, the concatenation symbol is slightly misleading, as it could
be interpreted as elementwise multiplication. Maybe replace it by $[\cdot,\cdot]$.


#### Post-rebuttal comments:
The authors' comments addressed my concerns on method and experimental details mostly well. I keep with my rating "6: Marginally above acceptance threshold".

---

> ### Author Response · Authors · 2020-11-25
> **Response to review (part 1/2)**
>
> Thanks a lot for the detailed feedback! We are very grateful that you found our idea novel and interesting. Below we address your concerns.
>
> (1) Training details
>
> Thanks for bringing up this issue! We apologize for not including sufficient details on training. We have updated Appendix A.1 with a new paragraph talking about how the components are trained in detail. We have also added some descriptions of the detection module in Section 3.1.
> To briefly summarize here, HyperDynamics for pushing is trained to optimize the sum of three losses: 1) the main dynamics prediction loss discussed in Equation (1). 2) object detection loss which follows the exact same form described in Mask R-CNN, as GRNNs follows Mask R-CNN for object detection, except that it operates in 3D now. It’s supervised using ground-truth object locations provided by the simulator. 3) a top-down view prediction loss using standard cross-entropy pixel matching loss, supervised with ground-truth top-down view of the object. All the components are trained jointly; none of them are pretrained. For loss 3, the training data alternates between the actual data collected during pushing, and data rendered by loading random objects from ShapeNet in the simulation. We do this because such rendered data can be easily and fast collected while actual pushing data is relatively expensive. Doing this allows us to train dynamics prediction only on a few objects (24 in our case), while letting our model to generalize to novel objects unseen during actual pushing.
>
> We have also move architecture and training details that were originally in appendix to the main paper now. (See implementation details in Section 4.1 and 4.2)
>
> (2) Moving from canonical to oriented shape representation.
>
>  Thanks for pointing out this! We apologize for causing the confusion about our orientation representation. To make it more clear, consider an object oriented at orientations A and B; we now have 2 design choices:
> - (1) treating them as the same object but with different orientation state, then we would feed its canonical shape into E_vis, and feed different orientations into the generated dynamics model (since the dynamics model takes as input the object state), or
> - (2) simply treating them as 2 different objects as if they have different shapes! (Imagine you consider a rotated mug as a new mug.) Then we can simply remove the orientation info from its state, only feed its position info (in addition to other system states) into the generated dynamics model, and encode their shape (which already includes their orientation) into the shape code z_vis.
>
> If we choose (1), then the problem is that when trained on only 24 objects, the model would only see 24 distinct z_vis, which could result in very bad overfitting for both E_vis and Hypernetwork H. So in our framework we chose (2), which enables both E_vis and H to see a much more smooth distribution of input data, and this helps our method to generalize. In practice, we did try (1) at first but the model failed to generalize.
>
> We hope this clarifies your doubt. We have also updated the last paragraph in Section 3.1 to make this design choice more clear.
>
> (3) Adding standard deviations in tables.
>
> Thank you for the suggestion! We have updated both Table 1 and 3 with standard deviations added. For table 2, it’s reporting success rate, so we believe its current form should be fine.
>
> (4) PlaNet listed as “no adaptation”
>
> Thanks for pointing out this! Indeed, PlaNet uses a recurrent state space model.  The problem that we are considering in the scope of this paper is potentially changing dynamics and physical properties of a dynamical system with a fixed set of properties. In PlaNet, the recurrent state representation is not designed with the explicit purpose to implicitly encode such property changes. However, it does present an experiment where only one agent is trained and it needs to infer which task it is facing by encoding past observations, while the dynamical system completely changes (even its morphology) from task to task. Therefore, we agree that it is a general framework that is able to handle changes in system dynamics. We have corrected this and moved PlaNet to group (ii) (Visual  dynamics  and  recurrent  state  representations). (Note that we have re-structured the introduction and the related work section following suggestions given by R1, and the groupings of the contemporary methods are now in Section 2.1.)

---

> > ### Author Response · Authors · 2020-11-25
> > **Response to review (part 2/2)**
> >
> > (5) Better performance on novel environments in Cheetah-Slope and Ant-Slope.
> >
> > This is because the novel environment is sampled from two ranges, one with steeper slope and one with more gentle slope than the slope range used for training. For the steeper slope range, the agent generalizes well and can run reasonably stable. For the more gentle slope, it’s actually an easier task and the agent can run very stably and go further than on slopes used in training. As a result, during testing time the average return can potentially be higher on novel environments than on seen environments. This doesn’t happen for Expert-Ens, since we believe the environment-specific experts have difficulty in handling those steeper slopes.
> >
> > (6) Typos and advices
> >
> > We really appreciate your help in finding all these typos! We have corrected them. Regarding the concatenation symbol in Figure 1, we have removed it from the figure and we believe it’s more clear now.
> >
> > Thanks again for the detailed comments and suggestions! We hope the revised paper explains the details of our approach better.

---

### Official Review · AnonReviewer3 · 2020-11-07
**Interesting work, but lacking many evaluation details**

**Rating:** 6
**Confidence:** 4

**Review:**

=== Summary

This paper proposes a framework, HyperDynamics, that takes in observations of how the environment changes when applying rounds of interactions, and then, generates parameters to help a learning-based dynamics model quickly adapt to new environments.

The framework consists of three modules:
- an encoding module that maps the observation of a few agent-environment interactions into a latent feature vector,
- a hypernetwork that conditions on the latent vector and generates all parameters of a dynamics model dedicated to the observed system, and
- a target dynamics model constructed using the generated parameters that predicts the future state by taking the current system state and the input action as input.

The authors evaluate the framework in a series of object pushing and locomotion tasks. They have shown that a single HyerDynamics model allows few-shot adaptation to new environments, outperforming several baselines while maintaining a performance that is on par with a set of models trained separately for each environment.


=== Strengths

This paper targets an important question of building a more generalizable dynamics model that can perform online adaptation to environments with different physical properties and scenarios that are not seen during training.

The authors have evaluated the method in several object pushing and robot locomotion tasks and shown superior performance over baselines that uses recurrent state representations or gradient-based meta-optimization.

Many practical treatments used in the pipeline can be good references for the community to learn from, e.g., how to encode object information in 3d, specific representation of the object orientation, and the use of Geometry-Aware Recurrent Networks (GRNNs) to learn 3D feature grids, etc.


=== Weaknesses

Although I like the idea of this paper, I believe the authors should provide more clarification and illustration of the experimental results to solidify the claims in the paper:

(1) What are the objects used in the pushing task? The authors claim that their "dataset consists of only 31 different object meshes with distinct shapes." It is important to include images of the objects to give the readers a better understanding of how diverse the dataset is and how different the geometry of the "seen" and "novel" objects are. This can help the readers better appreciate the generalization ability of the proposed method.

(2) It would be great if the authors can include some qualitative examples, e.g., video, to show the performance of the method. Purely from the numbers in the tables, it is hard for the readers to imagine how well the proposed approach solves the tasks.

(3) It would make the paper more illustrative if the authors can include some analysis and visualization of the learned representations in the middle of the network. For example:
- How are the latent embeddings different for different objects?
- Are there any correlations between the embeddings and the actual physical properties?
- How do the interactions affect the embedding? Will different interaction sequences result in the same embedding?
- How do the learned representations from Geometry-Aware Recurrent Networks (GRNNs) look like? The authors claim that it can "complete missing or occluded shape information." Can the authors provide some concrete evidence supporting this claim in the specific scenarios used in this paper? How do different numbers of interactions affect the quality of the representation?

(4) How does E_vis detect the objects in the scene? Are these detections in 2d or 3d? How accurate is the detection algorithm?

(5) The beginning of Section 3.1 describes that an object's orientation is represented as a quaternion. However, at the end of Section 3.1, the authors suggest that they "discard the orientation information from states fed into the generated dynamics model." This seems to me makes the "state" an incomplete representation of the environment, where the authors only predict the position of the object, which makes me wonder:

How does the model encode the geometry of the object? Will the missing of the orientation information introduce any ambiguities or uncertainties? What if the object is re-oriented? It may be better to include comparisons of different state representations.

Also, in Section 3.3, the authors suggest that they update the orientation using quaternion composition, which seems to be inconsistent with what has been described before. Haven't the model already discarded the orientation information?


=== Other comments

This paper only shows experiments in the simulation. I'm curious, are there any gaps before applying the method to the real world, and what are these gaps? For example, how long does the model take to optimize the action trajectories when performing MPC? Can it support real-time feedback control in real physical scenarios, especially when the environment is dynamic?


Model-predictive control relies on the environment's feedback to correct the action sequences, which can achieve a good control performance while tolerating a larger long-term prediction error. In your experiments, how important is the accuracy of the dynamics model? In other words, even if some baselines have a poorer forward prediction performance, will MPC be able to bridge some of the performance gaps?


In table 3, why are there multiple red numbers in the Ant-Slope columns?


Typo?
Page 4, Section 3.1:
"E_int then maps z_int to a 1-dimensional code z_int \in R^2."
This sentence seems weird. How does E_int map z_int to its own? Why does a "1-dimensional code" lies in a 2d space?


=== Post rebuttal

The authors' response and the revisions to the manuscript have greatly improved the quality and clarity of the paper. Most of my major concerns regarding the implementation and evaluation details have been sufficiently addressed; hence, I decide to increase the score from 5 (Marginally below acceptance threshold) to 6 (Marginally above acceptance threshold).

---

> ### Author Response · Authors · 2020-11-25
> **Response to review (part 1/3)**
>
> Thank you for your detailed comments and suggestions!  We are very encouraged that you found the problem we study important and our idea interesting. Below we address your concerns.
>
> (1) Details of objects used.
>
> Yes, we agree that including images of the objects would help readers to better understand how the selected objects are like. We have added a figure into Appendix A.1 showing all the objects used and the train-test split. We also added a sentence in the experimental section for pushing (section 4.1) in the main paper to refer readers to Appendix A. We hope it’s more clear now.
>
> (2) Qualitative example. E.g. video.
>
> Thanks for the suggestion! We agree that providing more qualitative examples such as a video would greatly help readers understand the capability of our methods. We are currently in progress of constructing a project website and also making a detailed video. We will make them publicly available once they are done.
>
> (3) Analysis and visualization of the learned latent representation.
>
> We now have included additional visualizations and analysis in Appendix B.4.  For the interaction encoder and the latent code z_int, we visualize the learned latent space in Figure 4 with varying number of interactions. We believe such additional results help answer your questions: indeed, there’s a correlation between the embeddings and the actual properties, and different interaction sequences would result in close embedding as long as their corresponding physical properties are close. The figure also suggests that the values of k we used are sufficient for the latent space to resemble the original space of physical properties well.
>
> As for the visual encoder E_vis, it is directly built upon Geometry-Aware Recurrent Networks (GRNNs) [1], and we would like to argue that the details of GRNNs themselves are not the main focus of this work, and we don’t consider deploying GRNNs in object pushing as one of our major contributions. We hope it’s reasonable to refer readers to the original paper [1] and its follow-up work [4, 5] for the latent representation visualization and details for shape completion. However, we do agree that providing some visualizations would help appreciate how the visual model works in our specific scenarios. We have added Figure 5 in Appendix B.4 with some qualitative results to show how our model learns to complete missing shape information and predicts the top-down view under potential occlusions.
>
> Regarding your last question on “how do different numbers of interactions affect the quality of the representation?”: The number of interactions only affects the produced interaction code z_int, as shown in Figure 4, while z_vis (produced by the visual encoder) is a function of the image input only and is not affected by the interactions.
>
> (4) How does E_vis detect objects?
>
> Our perception module E_vis directly uses GRNNs [1], which allow us to do both object detection and top-down view prediction. GRNNs detect objects using a 3D object detector that operates over its latent 3D feature map of the scene M. The object detector maps the scene feature map M to a variable number of axis-aligned 3D bounding boxes of the objects (and their segmentation masks if needed). Its architecture is similar to Mask R-CNN but instead uses RGB-D inputs and produces 3D bounding boxes. During our application, we crop the scene feature map using the object bounding boxes produced by GRNNs to obtain object-centric feature maps for shape encoding. We refer readers to [1] for the details of the detection pipeline. We do agree that adding some descriptions of it would make our paper more clear, so we have added some descriptions of the detection module in Section 3.1.
>
> The accuracy of this detection module is also studied and reported in [1]. We believe explicitly evaluating the accuracy of such 3D detector is out of the scope of this paper since neither 3D detection nor using GRNNs for object pushing is within the contributions of our work. In our experiments, since the baselines used for pushing (XYZ, Direct, MB-MAML and Expert-Ens) all use the same detector to obtain object positions (note that Expert-Ens only assumes access to ground-truth orientation, mass, and friction), we believe such comparison is fair and reasonable. However, since detection is a key stage in our dynamics prediction pipeline, we do agree that it’s good to present clear ablations to give readers an idea of how well such detector works. We have added a comparison between using ground-truth object positions and using detected object positions in Section B.1. The results show that the dynamics prediction performance of our model using detected object positions is close to the one using gt object positions, suggesting that the 3D detector in our pipeline works reasonably well.

---

> > ### Author Response · Authors · 2020-11-25
> > **Response to review (part 2/3)**
> >
> > (5) Orientation representation
> >
> >  Thanks for pointing out this! We apologize for causing the confusion about our orientation representation. Yes, a complete state of the object includes both position and orientation, and in fact the expert models in Expert-Ens use this complete state input. For HyperDynamics, essentially we consider object orientation as part of its shape information, instead of its state. To make it more clear, consider an object oriented at orientations A and B; we now have 2 design choices:
> > -  (1) treating them as the same object but with different orientation state, then we would feed its canonical shape into E_vis, and feed different orientations into the generated dynamics model (since the dynamics model takes as input the object state), or
> > -  (2) simply treating them as 2 different objects as if they have different shapes! (Imagine you consider a rotated mug as a new mug.) Then we can simply remove the orientation info from its state, only feed its position info (in addition to other system states) into the generated dynamics model, and encode their shape (which already includes their orientation) into the shape code z_vis.
> >
> > If we choose (1), then the problem is that when trained on only 24 objects, the model would only see 24 distinct z_vis, which could result in very bad overfitting for both E_vis and Hypernetwork H. So in our framework we chose (2), which enables both E_vis and H to see a much more smooth distribution of input data, and this helps our method to generalize. In practice, we did try (1) at first but the model failed to generalize.
> >
> > To answer your question ‘How does the model encode the geometry of the object?’: our E_vis, which is essentially GRNNs, have both an object detector and a shape encoder. The shape encoder learns to encode geometry via shape completion (top-down view prediction), and is also optimized jointly with other modules for motion prediction.
> >
> > Regarding questions about missing orientation info, re-oriented objects and quaternion composition: when predicting motion, we do predict both delta position and delta orientation. For unrolling, at any timestep t, we define the current orientation of object as (0,0,0,1) (quaternion without any rotation), and compose it with the predicted delta quaternion to obtain its quaternion for next timestep t+1. After that, we reorient the object feature map (obtained at timestep t) using this predicted quaternion and treat it as a new object to feed into E_vis, which produces z_vis and feeds into H. Therefore, this updated orientation is encoded in the weights of the generated dynamics model, and not fed as input to it. So essentially we are not ‘missing’ or ‘discarding’ any orientation information, but simply moving it from state input to shape input.
> >
> > We have updated the last paragraph in Section 3.1 to make this design choice more clear.
> >
> > (6) Applying in real world.
> >
> > Real-world application is indeed a very important topic in robotics research. Many methods have difficulty in transferring directly to real world sceneario mainly because the data distribution seen in real world differ significantly from the simulated data, especially the visual inputs. Our method is shown to generalize better to unseen environment in simulation. As for real world sceneraios, a very recent paper [2] also proposes to use GRNNs as the perception frontend to learn 3D object motion prediction under pushing, which enables viewpoint invariance of the learned dynamics model and shows good performance in real world when applying their model trained solely in simulation. (Their main contribution is to build a graph network over object-centric feature maps, while we focus on leveraging hypernetworks for generalization). The authors claim that their model’s good sim-to-real transferrability is achieved due to the fact that geometry information is shared by simulation and the real world by a large extent, and their models uses a 3D geometry-aware representation produced by GRNNs. Hyperdynamics for pushing also posseses such properties as it uses the same geometry-aware visual encoder, so we expect it to produce reasonable performance when transferred to real world. As for locomotion, our experiments follow the same setup as the MB-MAML [3] baseline, and uses the exact same input data. The locomotion task setting does not rely on visual input, so the mismatch between simulation and the real world should be less significant compared to the pushing setting.  In [3] the authors trained their method in real-world and claimed good performance. In our experimental section we do show that HyperDynamics for locomotion performs better than MB-MAML to unseen environments in simulation, and we see no major bottlenecks for collecting data and training our method in the real world following the same way. Therefore, we expect it to also produce a good (if not better) performance in a real world locomotion setting.

---

> > > ### Author Response · Authors · 2020-11-25
> > > **Response to review (part 3/3)**
> > >
> > > (6) Applying in real world. (cont.)
> > >
> > > That said, our main focus in this work is to evaluate how leveraging the multiplicative interactions provided hypernetworks can help improve performance and generalize to unseen environment properties in dynamics learning, and real-world application or sim-to-real transfer are not within the scope of this work. Another difficulty for such evaluation is that we do not have access to either a real robotic arm or the legged robot used in [3] due to the global pandemic. However, we do believe it’s a very interesting avenue for future work and we will try to evaluate real-world tasks in the future.
> > >
> > > As for the computation time required for performing MPC: 1) for pushing, the major inference cost is 3D object detection and shape encoding. In our current setting (MPC using random shooting with 30 action sequences), it takes ~1s for the model to take one pushing action, which is close to the length of a pushing action (800ms, 1 timestep used for pushing). Running the model in real-world for inference is OK, but collecting data in real world with such speed is indeed impractical. However, as discussed above, direct sim-to-real transfer is expected to perform reasonably well in real world, without any need for further real-world data collection, and such inference speed is reasonable for deploying the model in real-world. 2) for locomotion, the inference time of the network is much faster than the one used for pushing, since it does not rely on a 3D perception module. In our current setting, we evaluate 500 random action sequences for MPC and each step takes a few milliseconds, which is shorter than one timestep (10ms) defined for the task. As a result, we believe it does support real-time feedback control in a dynamic environment. Note that this is also shown in [3], and our method has an even faster inference speed since it does not need online gradient-based model update, as required by [3]. Therefore, we believe computation time is not an issue for applying the model in real world for locomotion.
> > >
> > > (7) MPC can bridge the performance gap.
> > >
> > > Yes, with the help of MPC indeed we can avoid error compounding especially in the case of long horizon tasks. However, MPC is not everything: it cannot help much if a model gives poor prediction performance. Thus, the prediction accuracy by itself is very important. In our experiments, we use the exact same MPC setting across all models, and we did show that our model outperforms the baselines in both prediction accuracy and control tasks with MPC. In fact, we tried to vary the number of action sequences evaluated and choose different planning horizons for both pushing and locomotion tasks, but our model outperforms the baselines consistently. Also, the current setting is nearly optimal: increasing those numbers further does not provide any further performance gain for any baseline.
> > >
> > > (8) Table 3 has multiple red numbers.
> > >
> > > Red numbers denote the best performance and the black ones represent the oracle performance (we mentioned this in section 4.2). For the Ant-Slope task, the recurrent baseline also performs well (on par with HyperDynamics), so the numbers are also marked in red. We have added one sentence to the last paragraph of section 4.2 to mention this.
> > >
> > > (9) Typo in Section 3.1.
> > >
> > > Thanks for pointing this out! Yes, it’s indeed a typo, and we actually meant ‘map \tau to z_int’. We have updated the sentence now.
> > > We used the phrase ‘1-dimensional code’ to suggest that it’s a vector. We agree that this is misleading and have removed this phrase.
> > >
> > > Thank you again for your constructive comments! Many of the suggestions are indeed very helpful. We have updated the manuscript and the supplementary materials based on your feedback.
> > >
> > >
> > >
> > > [1] Hsiao-Yu Fish Tung, Ricson Cheng, and Katerina Fragkiadaki. Learning spatial common sense with geometry-aware recurrent networks. In Proceedings of the IEEE Conference on Computer Vision and Pattern Recognition, pp. 2595–2603, 2019.
> > >
> > > [2] Hsiao-Yu Fish Tung, Zhou Xian, Mihir Prabhudesai, Shamit Lal, Katerina Fragkiadaki. 3D-OES: Viewpoint-Invariant Object-Factorized Environment Simulators. the Conference on Robot Learning (CoRL), 2020.
> > >
> > > [3] Anusha Nagabandi, Ignasi Clavera, Simin Liu, Ronald S. Fearing, Pieter Abbeel, Sergey Levine, and Chelsea Finn. Learning to adapt in dynamic, real-world environments through metareinforcement learning. In International Conference on Learning Representations, 2019.
> > >
> > > [4] Adam W. Harley, Fangyu Li, Shrinidhi K. Lakshmikanth, Xian Zhou, Hsiao-Yu Fish Tung, Katerina Fragkiadaki. Learning from Unlabelled Videos Using Contrastive Predictive Neural 3D Mapping. In International Conference on Learning Representations, 2020.
> > >
> > > [5] http://www.cs.cmu.edu/~aharley/viewcontrast/

---

### Author Response · Authors · 2020-11-25
**General Response**

We thank all the reviewers for their detailed feedback and useful suggestions! We have tried our best to address all the concerns of the reviewers in the individual responses. We have also updated our manuscript with requested experiments and visualizations. We summarize the major changes below:

- In Appendix A.1, we added an image and descriptions of the objects we used for data collection. (R3, R1)
- In Appendix B.4, we added visualizations of the learned latent space of the interaction encoder E_int with different values of k, and qualitative examples of the predicted top-down view from the visual encoder E_vis. (R3, R1)
- In Appendix B.1, we added an ablation study of our model to illustrate the effect of the 3D detector, the visual decoder, the cropping step, and the dimensions of the latent code. (R3, R2, R1)
- In Appendix A.1, we added a paragraph about the details of our model architecture, especially the visual encoder built upon GRNNs, and its training details. (R4, R1)
- We moved architecture and training details that were originally in appendix to the main paper now, and made the details of the baselines more clear. (See implementation details in Section 4.1 and 4.2) (R4, R2, R1)
- In Table 1 and 3, we added standard deviation values. We also included standard deviations in all the newly added tables. (R4)
- In Figure 1, we removed the misleading concatenation symbol. (R4)
- In Appendix B.2, we added an analysis on model capacity. (R2)
- In Appendix B.3, we added additional comparison on prediction errors for all the models on the locomotion tasks. (R2)
- We restructured the introduction and related work sections, to make the paper more succinct and reader friendly. We also summarized our contributions with bullet points at the end of the introduction. (R1)

---

### Decision · Program_Chairs · 2021-01-07
**Final Decision**

**Decision:**

Accept (Poster)

**Comment:**

This paper proposes "HyperDynamics" a framework that takes into account the history of an agents recent interactions with the environment to predict physical parameters such as mass and friction. These parameters are fed into a forward dynamics model, represented as a neural network, that is used for control.

Pros:
- addresses an important problem (adapting dynamics models to "new" environments) and provides strong baselines
- well written and authors have improved clarity even further based on reviewers comments

Cons:
- I agree with the reviewer that it is currently unclear how well this will transfer to the real world
- The idea of predicting physical parameters from a history of environment interactions is not not novel in itself (although the proposed framework is, as far as I know). The authors should include related work along the lines of (1) (this is just one paper that comes to mind, others exist)

(1) Preparing for the Unknown: Learning a Universal Policy with Online System Identification